# Universal Multiclass Transductive Online Learning

**Steve Hanneke** [1]  **Hongao Wang** [1]

## Abstract

We consider the problem of universal transductive online classification with a possibly unbounded label space. This setting considers online learning, with the sequence of instances (without labels) known to the learner in advance. We say a concept class $\mathcal{H}$ is learnable if there is a learning algorithm $\mathcal{A}$, such that for every realizable sequence, the number of mistakes made by $\mathcal{A}$ grows at most sublinearly with the number of predictions. We characterize the learnability of this setting and show that there are only two possible optimal rates for the learnable classes: either bounded or increasing logarithmically. We introduce a new combinatorial structure, called "Level-Constrained-Littlestone-Littlestone (LCLL) tree", which, along with the *indifference* property, characterizes the learnability. We also extend the learnability result to the agnostic case and the case where only the stochastic process that generates the instance sequence is known.

## 1. Introduction

Online learning (Littlestone, 1988) is a sequential game. At each round, the adversary chooses an instance from the instance space $\mathcal{X}$. After the learner makes its prediction, the adversary chooses the true label from the label space $\mathcal{Y}$. The goal is to minimize the number of mistakes made by the learner. Under this setting, the learner has to face two types of uncertainties: *labeling-related* uncertainty and *instance-related* uncertainty. It is an interesting question to figure out what specific role each type of uncertainties, especially the labeling-related uncertainty, plays in the complete picture. Therefore, Ben-David et al. (1997) introduced a new learning model to remove the instance-related uncertainty, which is called offline learning. In that model, the adversary chooses the instance sequence but reveals it to the learner

in advance. Recently, Hanneke et al. (2023b) renamed this setting to transductive online learning, due to the similarity to transductive PAC learning (Vapnik, 2006). Both focus on investigating the benefits of knowing the unlabeled data beforehand.

Another motivation for investigating the universal transductive online learning is from the work of Hanneke & Wang (2024). They investigated the problem of optimistically universal online learning with general concept classes for binary classification. In that work, they introduced the assumption of concept classes into the model of learning under minimal assumptions on the stochastic process that generates the instance sequence. The assumption on the stochastic process they made is that online learning is possible if the learner knows that stochastic process, which is called "the process admits universal online learning". Therefore, the problem of when all processes admit universal online learning is equivalent to the universal online learnability problem, where only the stochastic process generating the instance sequence is known. This is closely related to the universal transductive online learning problem, which is the universal online learnability problem, where the instance sequence is known.

**Transductive Online Learning.** Just as online learning, transductive online learning is a sequential game as well. There are two players, the adversary and the learner. Before the game starts, the adversary may pick a sequence of instances $\mathbb{X} = \{X_t\}_{t \in \mathbb{N}}$, so that for every $t$, $X_t \in \mathcal{X}$, where $\mathcal{X}$ is a non-empty set called instance space. The instance sequence $\mathbb{X}$ is revealed to the learner in advance. Then at each round $t$, the learner makes a prediction $\hat{Y}_t$ based on the instance sequence $\mathbb{X}$ and the history of the true label $Y_{<t} = \{Y_i\}_{i<t}$. Then the adversary reveals the true label $Y_t \in \mathcal{Y}$, which can be used to inform the future prediction. Here $\mathcal{Y}$ is a countable set called label space. The learner suffers a loss, defined by the indicator function $\mathbb{I}[\hat{Y}_t \neq Y_t]$. For any learning algorithm, we use the number of mistakes to measure its performance, which is $\sum_{t=1}^{T} \mathbb{I}[\hat{Y}_t \neq Y_t]$, in the realizable setting.

This model was first introduced in the work of Ben-David, Kushilevitz, and Mansour (1997). Their main goal is to figure out how the label-related uncertainty affects the online learning procedure. Recently, the work of Hanneke

[1] Department of Computer Science, Purdue University, West Lafayette, IN 47907, USA.. Correspondence to: Hongao Wang <wang5270@purdue.edu>.

*Proceedings of the 43$^{rd}$ International Conference on Machine Learning*, Seoul, South Korea. PMLR 306, 2026. Copyright 2026 by the author(s).

et al. (2023b) proved a trichotomy and showed that transductive online binary classification only has three possible rates $O(1)$, $\Theta(\log T)$, and $\Omega(T)$. They proved that if the Littlestone dimension of $\mathcal{H}$ is finite, there is an algorithm that only makes finite mistakes to learn the target concept. If $\mathcal{H}$ has finite VC dimension but has an infinite Littlestone dimension, the best rate is $\log T$. And if $\mathcal{H}$ has infinite VC dimension, there is an adversary that can push any algorithm to make a mistake every round. More recently, the work of Hanneke et al. (2024b) extended the result from binary classification to multiclass classification with countably infinite label spaces. In that work, they also proved a trichotomy. There are only three types of different rates, which are $O(1)$, $\Theta(\log T)$, and $\Omega(T)$. And the characterizations are the level-constrained branching dimension and the level-constrained Littlestone dimension. There is a substantial amount of literature investigating this topic recently, such as Hanneke & Shaeiri (2025); Chase et al. (2025). However, all of the previous works focus on the uniform rates of the transductive online learning model. In other words, the bound is independent of the sequences.

**Universal Learning.** In the work of Bousquet et al. (2021), they considered the distribution-dependent learning rates and sequence-dependent learning rates, in PAC learning and online learning, respectively, which is called "universal learning rates". We focus on the universal rates of online learning here. Unlike uniform rates, which are sequence-independent, the universal rates are described as follows: for every realizable sequence, there is a function that bounds the number of mistakes. This is a more realistic measure of learning rate, as during one learning procedure, we only need to face one specific data sequence. There are many works investigating the performance of online learning algorithms under the universal setting, such as (Blanchard et al., 2022; Blanchard, 2022; Hanneke et al., 2023c; 2025). There is also a line of work investigating the universal PAC learning, such as Hanneke et al. (2022; 2023c); Hanneke & Xu (2024); Hanneke et al. (2024a).

**Multiclass Learning.** Daniely et al. (2015) extended the online learning from binary classification to multiclass classification with unbounded label spaces and showed that the Littlestone dimension characterizes the online learnability in the realizable case. Hanneke et al. (2023a) show that the Littlestone dimension also characterizes the online learnability in the agnostic setting. We would like to discuss the cases in which the label space is unbounded, because many interesting scenarios happen when we try to extend the label space from finite to unbounded. In PAC learning, there is also a line of work extending the binary classification to multiclass classification (Natarajan & Tadepalli, 1988; Natarajan, 1989; Brukhim et al., 2022).

In this paper, we investigate the universal learning rates of transductive online multiclass learning with a countably infinite label space.[1] For the realizable setting, we provide a trichotomy for the universal multiclass transductive online learning. To describe this trichotomy, we use two combinatorial structures: the indifferent Littlestone tree and the indifferent Level-Constrained-Littlestone-Littlestone (LCLL) tree, which will be defined later. Informally, the trichotomy can be stated as follows.

- If $\mathcal{H}$ has no infinite indifferent Littlestone tree, it can be learned with a constant number of mistakes.
- If $\mathcal{H}$ has an infinite indifferent Littlestone tree but has no infinite indifferent LCLL tree, it can be learned with $\Theta(\log T)$ number of mistakes.
- If $\mathcal{H}$ has an infinite indifferent LCLL tree, it cannot be learned with $o(T)$ number of mistakes.

We also extend the learnability results to the agnostic case and provide a $\tilde{O}(\sqrt{T})$ upper bound for the concept classes with no infinite indifferent LCLL tree. $\tilde{O}(\sqrt{T})$ hides the poly-logarithmic factors.

For binary classification, the results of Hanneke & Wang (2024) imply that the VCL tree characterizes the universal transductive online learnability. Therefore, several combinatorial structures that extend the VCL tree to the multiclass setting may characterize the universal multiclass transductive online learnability, such as the DSL tree (Hanneke et al., 2023c), the LCL tree (Hanneke et al., 2024b), or the LCL-Littlestone tree. Surprisingly, it turns out that none of these choices is the right answer.

Recall the proof of the universal transductive online learnability for binary classification. They use the VCL game, which is a Gale-Stewart game[2], defined in Bousquet et al. (2021) to prove the upper bound. Then, due to a lemma in Bousquet et al. (2023), every concept class that has an infinite Littlestone tree (VCL tree) also has an infinite indifferent Littlestone tree (VCL tree). They say that the concept has an indifferent Littlestone tree if, for every node in the tree, all its descendants agree on the value of the instances that precede it in Breadth-First-Search order. This property allows the adversary to choose the true labels by performing a random walk starting from the root, even though the learner knows the instance sequence in advance.

However, this nice lemma does not hold when label spaces are unbounded. In fact, there exists a concept class $\mathcal{H}$ that has an infinite Littlestone (LCLL) tree but has no infinite indifferent Littlestone (LCLL) tree, shown in Example 3.1. This example also suggests that the indifferent Littlestone

---

[1]The results for the realizable setting can be extended to uncountable label spaces, but not in the agnostic setting, as we require the number of experts to be countable.

[2]A Gale-Stewart game is a perfect information two-player infinite game. More details are provided in Appendix A.

(LCLL) tree should be the correct characterization. However, the structure of the Gale-Stewart game is naturally related to the Littlestone-tree type of trees and is difficult to apply to indifferent trees. At each round of the game, the adversary proposes a sequence of instances, and the learner picks a sequence of labels. Therefore, the winning strategy of the adversary automatically leads to an infinite Littlestone-type tree. This mismatch prompts us to develop a new method for designing a Gale-Stewart game that captures the indifferent property of the tree. It turns out that the adversary needs to provide all the possible label sequences for an interval of instances, i.e., the adversary needs to provide all possible labels for instances from $X_{t_k+1}$ to $X_{t_{k+1}}$ such that $X_{t_{k+1}}$ has two labels[3], instead of a sequence of instances. As far as we know, it is the first time to design such a type of Gale-Stewart game in the universal learning literature, and this design would be useful to solve any problem whose characterization is the infinite indifferent Littlestone-type trees. We want to highlight this design of the Gale-Stewart game as the main conceptual contribution of this work, and we believe it can be used as a tool for more multiclass learning problems in the online learning context.

As we mentioned before, the universal transductive online learnability problem is closely related to the characterization of the case where all processes admit universal online learning. Therefore, we also extend the learnability result to the case where only the stochastic process generating the instance sequence is known in advance, that is, the condition that all processes admit universal multiclass online learning.

**Organization of the paper.** In this paper, we first provide the notations and the main results in section 2. Then we provide several examples to show that some reasonable guesses are not the correct characterization in section 3. After that, we show the high-level ideas of the proof of the trichotomy and the learnability results in Section 4 and Section 5. Due to the lack of space, we put our detailed proofs and the results for stochastic processes in the appendices.

## 2. Preliminaries and Main Results

In this section, we provide formal definitions, model settings, and a brief list of our main results.

**Model Setting.** We formally provide our learning model here. $\mathcal{X}$ and $\mathcal{Y}$ are both non-empty sets. $\mathcal{X}$ is the instance space and $\mathcal{Y}$ is the label space. Here we focus on the learning on 0-1 loss: that is, $(y, y') \mapsto \mathbb{I}[y \neq y']$, where $\mathbb{I}[\cdot]$ is the indicator function. The concept class $\mathcal{H} \subseteq \mathcal{Y}^{\mathcal{X}}$ is a non-empty set of measurable functions from $\mathcal{X}$ to $\mathcal{Y}$. The instance sequence $\mathbb{X} = \{X_t\}_{t \in \mathbb{N}}$ is a sequence of elements $X_t \in \mathcal{X}$, and the label sequence $\mathbb{Y} = \{Y_t\}_{t \in \mathbb{N}}$ is a sequence

---

[3]Here we use the Littlestone tree as an example, as it is easier to describe than LCLL tree.

of elements $Y_t \in \mathcal{Y}$. In this paper, we often use $(X_{\leq t}, Y_{\leq t})$ to stand for $\{(X_i, Y_i)\}_{i \leq t}$.

We also use the definition of a partial concept, which is a partial function from $\mathcal{X}$ to $\mathcal{Y}$. A partial function allows the label of some instances to be undefined, and no matter what prediction was made on that instance, it is a mistake.

In the transductive online learning setting, the instance sequence and the label sequence are both chosen by the adversary. However, the instance sequence $\mathbb{X}$ is revealed to the learner in advance, so that the learner can leverage the information of future instances to help design the learning algorithm.

The transductive online learning algorithm is a sequence of measurable functions $f_t : \mathcal{X}^{\infty} \times \mathcal{Y}^{t-1} \to \mathcal{Y}$, where $t$ is a non-negative integer and it is usually represented by $\mathcal{A}$. For convenience, we usually use $\hat{Y}_t$ as the prediction of the algorithm $\mathcal{A}$ at round $t$.

Following the tradition of universal learning, we define the set of realizable sequences as follows.

**Definition 2.1.** For every concept class $\mathcal{H}$, we can define the following set of processes $\mathrm{R}(\mathcal{H})$:

$$\mathrm{R}(\mathcal{H}) := \{(\mathbb{X}, \mathbb{Y}) = \{(X_i, Y_i)\}_{i \in \mathbb{N}} :$$
$$\forall n < \infty, \{(X_i, Y_i)\}_{i \leq n} \text{ realizable by } \mathcal{H}\}.$$

Following the tradition of online learning, we use the number of mistakes to measure the performance of the online learning algorithm in the realizable case, which is defined as follows.

**Definition 2.2.** The number of mistakes of the learning algorithm $\mathcal{A}$ with respect to the data sequence $(\mathbb{X}, \mathbb{Y})$ is $M(\mathcal{A}, (\mathbb{X}, \mathbb{Y}), T) = \mathbb{E}[\sum_{t=1}^{T} \mathbb{I}[Y_t \neq \hat{Y}_t]]$.

However, in the agnostic case, we do not have the assumption that the label sequences are all realizable, instead, they may be arbitrary. Therefore, the number of mistakes made by the learning algorithm can be arbitrarily large and it is not an appropriate measure of learnability. Hence, we use *regret* to measure the learnability, which is the difference between the performance of our algorithm and the best realizable sequence. Formally,

**Definition 2.3.** The regret of the learning algorithm $\mathcal{A}$ with respect to the data sequence $(\mathbb{X}, \mathbb{Y}, \mathbb{Y}^*)$, where $(\mathbb{X}, \mathbb{Y}^*) \in \mathrm{R}(\mathcal{H})$ is $\mathrm{Regret}(\mathcal{A}, (\mathbb{X}, \mathbb{Y}, \mathbb{Y}^*), T) = \mathbb{E}[\sum_{t=1}^{T}(\mathbb{I}[Y_t \neq \hat{Y}_t] - \mathbb{I}[Y_t \neq Y_t^*])]$.

If there is an algorithm $\mathcal{A}$, such that $\mathrm{Regret}(\mathcal{A}, (\mathbb{X}, \mathbb{Y}, \mathbb{Y}^*), T) = o(T)$ for every sequence $(\mathbb{X}, \mathbb{Y}, \mathbb{Y}^*)$, where $(\mathbb{X}, \mathbb{Y}^*) \in \mathrm{R}(\mathcal{H})$, we say $\mathcal{H}$ is universally transductively online learnable for the agnostic case.

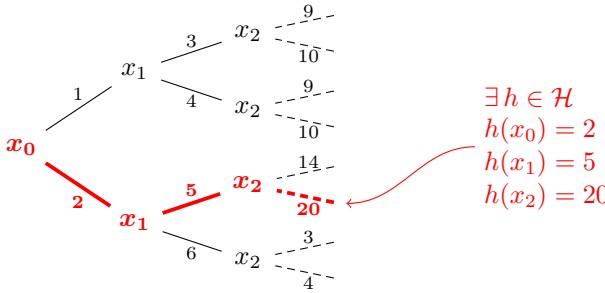

*Figure 1.* A Level-Constrained Littlestone tree of depth 3. Every branch is consistent with a concept $h \in \mathcal{H}$. We take $\mathcal{H} \subseteq \mathbb{N}^{\mathcal{X}}$ for convenience. The only restriction is that the two edges connecting two children of the same node should be labeled with different labels. This is illustrated here for one of the branches.

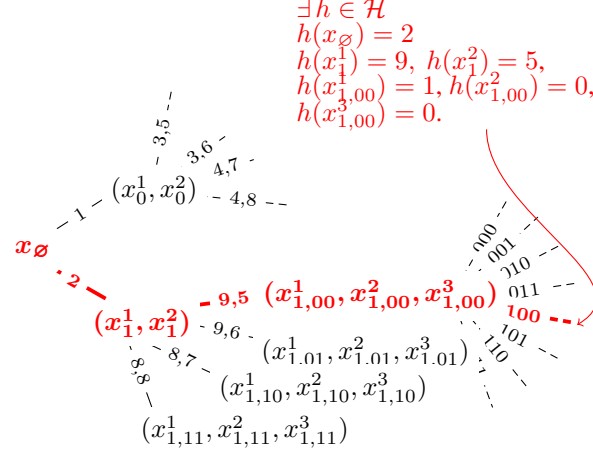

*Figure 2.* An LCLL tree of depth 3. Every branch is consistent with a concept $h \in \mathcal{H}$. This is illustrated here for one of the branches. Due to a lack of space, not all external edges are drawn. In this figure, we take $\mathcal{H} \subseteq \mathbb{N}^{\mathcal{X}}$ for convenience. This figure is modified from the work of Bousquet et al. (2021).

It is common and traditional to use combinatorial structures related to the concept class to build the mistake or regret bound in the online learning setting. We follow this tradition and provide the combinatorial structures we use here.

We first define the Littlestone tree for multiclass learning.

**Definition 2.4** (Littlestone Tree(Hanneke et al., 2023c))**.** A Littlestone Tree for $\mathcal{H}$ of depth $d \leq \infty$ is a collection

$$\{x_u \in \mathcal{X} : 0 \leq k < d, u \in \{0,1\}^k\}$$

and a sequence of functions $\mathbf{y}_i : \{0,1\}^i \to \mathcal{Y}$ satisfy

1. For every $i < d$, $\mathbf{y}_i(u_{<i}, 0) \neq \mathbf{y}_i(u_{<i}, 1)$, where $u_i \in \{0,1\}$ and $u_{<i} = (u_1, \ldots, u_{i-1})$.
2. There exists a concept $h \in \mathcal{H}$, such that for every $i \leq d$, $h(x_{u_{<i}}) = \mathbf{y}_i(u_{<i}, u_i)$, where $u_i \in \{0,1\}$ and $u_{<i} = (u_1, \ldots, u_{i-1})$.

Then we provide a combinatorial dimension used in the multiclass transductive online learning, an extension of VC dimension in the multiclass setting, which is called the Level-Constrained Littlestone dimension. Intuitively, it is a Littlestone tree, but every node in the same depth are labeled with the same instance. Check Figure 1 for illustration.

**Definition 2.5** (Level-Constrained Littlestone Tree (LCL tree))**.** A Level-Constrained Littlestone tree (LCL tree) for $\mathcal{H}$ is a complete binary tree of depth $d \leq \infty$ whose internal nodes in depth $k$ are labeled by element $x_k \in \mathcal{X}$, and whose two edges connecting a node to its children are labeled by $y_1, y_2 \in \mathcal{Y}$ ($y_1 \neq y_2$), such that every finite path emanating from the root is consistent with a concept $h \in \mathcal{H}$.

If $\mathcal{H}$ has an LCL tree with depth $d$, but doesn't have one with depth $d + 1$, we say the LCL dimension of $\mathcal{H}$ is $d$.

We then need to define the LCL-Littlestone tree, which is an extension of the VCL tree to multiclass learning. Intuitively, it is a Littlestone tree, but every node at depth $k$ is not labeled by an instance but by a sequence of $k + 1$ instances,

which can form a depth $k + 1$ LCL tree, where the leaves are not labeled. Each node has not just 2 children but $2^{k+1}$ children, and each edge is not labeled by a label but by a sequence of labels, which is a path from the root to one leaf in the depth-$k + 1$ LCL tree. Check Figure 2 for illustration.

**Definition 2.6** (LCL-Littlestone Tree)**.** An LCL-Littlestone tree for $\mathcal{H}$ of depth $d \leq \infty$ is a collection

$$\{x_u \in \mathcal{X}^{k+1} : \\ 0 \leq k < d, u \in \{0,1\}^1 \times \{0,1\}^2 \times \cdots \times \{0,1\}^k\}$$

and a sequence of functions $\mathbf{y}_{n,i} : \{0,1\}^1 \times \{0,1\}^2 \times \cdots \times \{0,1\}^n \times \{0,1\}^i \to \mathcal{Y}$ satisfy

1. $\mathbf{y}_{n,i}((u_{\leq n}, (u_{n+1}^{\leq i-1}, 0))) \neq \mathbf{y}_{n,i}((u_{\leq n}, (u_{n+1}^{\leq i-1}, 1)))$, for every $n < d$ and $i \leq n + 1$. Here $u_n \in \{0,1\}^n$, $u_{\leq n} = (u_1, u_2 \ldots, u_n)$ and $u_{n+1}^{\leq i-1} = (u_{n+1}^1, \ldots, u_{n+1}^{i-1})$.
2. There exists a concept $h \in \mathcal{H}$, such that $h(x_{u_{\leq k}}^i) = \mathbf{y}_{k,i}((u_{\leq k}, (u_{k+1}^1, \ldots, u_{k+1}^i)))$ for all $1 \leq i \leq k + 1$ and $0 \leq k \leq n$, where we denote

$$u_{\leq k} = (u_1^1, (u_2^1, u_2^2), \ldots, (u_k^1, \ldots, u_k^k)),$$
$$x_{u_{\leq k}} = (x_{u_{\leq k}}^1, \ldots, x_{u_{\leq k}}^{k+1})$$

We say that $\mathcal{H}$ has **an infinite LCL-Littlestone tree** if it has an LCLL tree of depth $d = \infty$.

Then we need to define the *indifferent* property of the tree, which can be applied to the Littlestone tree or the LCLL tree defined before. This definition is extended to unbounded label spaces from the same definition in the work of Bousquet et al. (2023), which only defines the property on binary label

spaces. Intuitively, this property says all the descendants of a node agree on the value of the node that comes before that node in Breadth-First-Search Order.

**Definition 2.7** (Indifferent Tree. Extended from the work of Bousquet et al. (2023)). Let $\mathcal{X}$ be a set and $\mathcal{H} \subseteq \mathcal{Y}^{\mathcal{X}}$ be a hypothesis class, and let

$$T = \{x_{\mathbf{u}} \in \mathcal{X} : \mathbf{u} \in (\{0,1\})^*\} \text{ and } \{\mathbf{y}_i\}_{i \in \mathbb{N} \cup \{0\}}$$

be an infinite perfect binary tree that is shattered by $\mathcal{H}$. This implies the existence of a collection

$$\mathcal{H}_T = \{h_{\mathbf{u}} \in \mathcal{H} : \mathbf{u} \in (\{0,1\})^*\}$$

of consistent functions.

We say such a collection is **_indifferent_** if for every $\mathbf{v}, \mathbf{u}, \mathbf{w} \in (\{0,1\})^*$, if $\mathbf{v}$ comes before $\mathbf{u}$ in Breadth-First-Search order, and $\mathbf{w}$ is a descendant of $\mathbf{u}$ in the tree $T$, then $h_{\mathbf{u}}(x_{\mathbf{v}}) = h_{\mathbf{w}}(x_{\mathbf{v}})$. In other words, the functions for all the descendants of a node that appears after $\mathbf{v}$ agree on $\mathbf{v}$. We say that $T$ is **_indifferent_** if it has a set $\mathcal{H}_T$ of consistent functions that are indifferent.

Then we can provide the trichotomy for the realizable case and the learnability result for the agnostic case.

**Theorem 2.8.** *For the realizable case, against any adversary (represented by $(\mathbb{X}, \mathbb{Y}) \in R(\mathcal{H})$), we have the following trichotomy:*

1. *If and only if $\mathcal{H}$ does not have an infinite indifferent Littlestone tree, there exists an algorithm $\mathcal{A}$, such that $M(\mathcal{A}, (\mathbb{X}, \mathbb{Y}), T) = O(1)$.*
2. *If and only if $\mathcal{H}$ has an infinite indifferent Littlestone tree but does not have an infinite indifferent LCLL tree, there exists an algorithm $\mathcal{A}$, such that $M(\mathcal{A}, (\mathbb{X}, \mathbb{Y}), T) = \Theta(\log T)$.*
3. *If $\mathcal{H}$ has an infinite indifferent LCLL tree, no learners can guarantee sublinear mistakes.*

**Theorem 2.9.** *A concept class $\mathcal{H}$ is agnostically universally transductively online learnable, if and only if $\mathcal{H}$ has no infinite indifferent LCLL tree.*

We also extend the learnability results to the case where only the stochastic process generating the instance sequence is known to the learner in advance. This result answers the condition when all processes admit universal online learning. We provide the details and discussion of this setting in the appendices.

## 3. Examples

In this section, we provide some insightful examples to show that several natural combinatorial structures are not the right choices for characterization.

**Example 3.1** ($\mathcal{H}$ with infinite LCLL tree, but learnable.). *Let $\mathcal{X}$ be the instance space, $\mathcal{Y} = \mathbb{N} \cup \{0\}$ be the label space. Let $u = (u_1^1, (u_2^1, u_2^2), \ldots, (u_k^1, \ldots, u_k^k)) \in \{0,1\} \times \{0,1\}^2 \times \cdots \times \{0,1\}^k$, $k \in \mathbb{N}$. Consider the following concept class:*

$$\mathcal{H} = \{h_u : \forall u, h_u(X_{u_{\leq k-1}}^i) = u_k^i,$$
$$\text{and all the other } X \in \mathcal{X}, h_u(X) = |u| + 1\},$$

*where* $u_{\leq k-1} = (u_1^1, (u_2^1, u_2^2), \ldots, (u_{k-1}^1, \ldots, u_{k-1}^{k-1}))$, $X_{u_{\leq k-1}} \in \mathcal{X}^k$, *and $|u|$ is the number of different tuples in $u$. For example, $|(u_1^1, (u_2^1, u_2^2), \ldots, (u_k^1, \ldots, u_k^k))| = k$.*

First, we want to claim that $\mathcal{H}$ has an infinite LCLL tree. It is easy to see that for the following collection:

$$T = \{X_u \in \mathcal{X}^{k+1} : u \in \{0,1\} \times \{0,1\}^2 \times \cdots \times \{0,1\}^k\}$$

and the sequence of functions $\mathbf{y}_{k,i}(u) = u_{k+1}^i$, for any $k$, any $u$, and any $j \leq k$, any $i \leq j+1$, $h_u(X_{u_{\leq j}}^i) = \mathbf{y}_{j,i}((u_{\leq j}, (u_j^1, \ldots, u_j^i)))$. That means there is an infinite LCLL tree shattered by the concept class $\mathcal{H}$.

However, for any fixed instance sequence $\mathbb{X}$, we can learn it with a finite number of mistakes by using the following method. For convenience, we can assume the target concept is $h_u$. If there is an $X_t \notin \{X_{u_{\leq k}}^i, \forall k \leq |u|, i \leq k+1\}$, by witnessing its label, we know $|u| + 1$. There will be only a finite number of different functions; thus, we can learn the target function by making finite mistakes. Then we know every $X_t \in \{X_{u_{\leq k}}^i, \forall k \leq |u|, i \leq k+1\}$, and we can make the prediction of $X_t$ by following $u$. Thus, this example demonstrates that a concept class with an infinite LCLL tree remains universally transductively online learnable.

*Remark* 3.2. We built this example very carefully to make sure we only use countably infinite labels. If we allow uncountable label space, we can change $|u| + 1$ by $u$.

This example also shows that the DSL tree is not the correct answer, as an infinite LCLL tree is also an infinite DSL tree. A similar construction shows that there is a concept class that has an infinite Littlestone tree but does not have an infinite indifferent Littlestone tree.

**Example 3.3** ($\mathcal{H}$ with no infinite LCL tree, but not learnable.(Bousquet et al., 2021)). *Let $\mathcal{X}$ be the instance space, $\mathcal{Y} = \{0,1\}$ be the label space. Let $u = (u_1^1, (u_2^1, u_2^2), \ldots, (u_k^1, \ldots, u_k^k)) \in \{0,1\} \times \{0,1\}^2 \times \cdots \times \{0,1\}^k$, $k \in \mathbb{N}$. Consider the following collection:*

$$T = \{X_u \in \mathcal{X}^{k+1} : u \in \{0,1\} \times \{0,1\}^2 \times \cdots \times \{0,1\}^k\}$$

*Then for every $u$, we define*

$$\mathcal{H}_u = \{h : h(X_{u_{\leq k}}^i) = u_{k+1}^i, h(X_u^i) = 0 \text{ or } 1,$$
$$h(X) = 1 \text{ for all the other } X.\}$$

*and $\mathcal{H} = \bigcup_u \mathcal{H}_u$. Here $u_{\leq k} = (u_1, u_2, \ldots, u_k)$.*

It is easy to notice that for every $d \in \mathbb{N}$, for every $k \leq d$, we have the collection $T$ which is shattered by $\mathcal{H}$. And for every node in the LCLL tree, all the functions for its descendants agree on the nodes that come before it in lexicographic order (labeled with 1). However, for any given sequence $\mathbb{X} = \{X_t\}_{t \in \mathbb{N}}$, as $X_1 \in X_u$ for some $u$, the depth of the LCL tree defined by this sequence is at most $|u| + 1$. Here $|u|$ is the number of different tuples in $u$. For example, $|(u_1^1, (u_2^1, u_2^2), \ldots, (u_k^1, \ldots, u_k^k))| = k$. Thus, $\mathcal{H}$ does not have an infinite LCL tree.

# 4. Realizable Setting

In this section, we focus on the realizable setting and provide the high-level idea of how to prove Theorem 2.8.

## 4.1. No Sublinear Mistake Bound

We first prove the lower bound when $\mathcal{H}$ has an infinite indifferent LCLL tree.

**Lemma 4.1.** *If $\mathcal{H}$ has an infinite indifferent LCLL tree, the adversary can choose a data sequence to force any learner to make more than sublinear mistakes.*

For brevity, we only provide a proof sketch here, and the complete proof is in Appendix B.

*Proof Sketch.* First, we can modify the indifferent infinite LCLL tree such that it has the property that the number of elements contained by the $k$-th node in the Breadth-First-Search (BFS) order is $2^{k-1}$. The instance sequence is all the instances that come in the lexical order in each node and the BFS order among different nodes. Then we take a random walk on this tree to choose the true label for each instance. The instances in the node visited by the random walk are labeled by the label on the edge adjacent to it in the path. The instances in an off-branch node are labeled by the label decided by its descendants. (We can do this as the tree is indifferent.) Thus, when reaching a node on the path, no matter what the algorithm predicts, it makes mistakes with probability $\frac{1}{2}$. Thus, it makes a quarter mistake in expectation. Then, by Fatou's lemma, for each learning algorithm, we get a realizable process such that the algorithm does not make a sublinear loss almost surely. $\square$

## 4.2. Logarithmic Mistake Bound

Then we show the $\log T$ upper bound, if $\mathcal{H}$ has no infinite indifferent LCLL tree.

**Lemma 4.2.** *If a concept class $\mathcal{H}$ does not have an infinite indifferent LCLL tree, there exists an algorithm $\mathcal{A}$, such that, for every $(\mathbb{X}, \mathbb{Y}) \in R(\mathcal{H})$, $M(\mathcal{A}, (\mathbb{X}, \mathbb{Y}), T) = O(\log T)$.*

To prove this lemma, we need to design a transductive on-line learning algorithm. First, we provide the following definition.

**Definition 4.3.** For a level-constrained tree, whose root and internal nodes are labeled by the elements of a sequence $\mathbb{X}$. Each node of the tree may have one child or two children. Then for every $k \in \mathbb{N}$, we say a subsequence $(X_{t_1}, \ldots, X_{t_k})$ is embedded in the level-constrained tree if:

1. There is a subtree, whose root is labeled by $X_{t_1}$ and has two children.
2. For every $1 \leq i \leq k$, subsequence $(X_{t_i}, \ldots, X_{t_k})$ is embedded in the subtree whose root is the child of $X_{t_{i-1}}$[4].

If the level-constrained tree is shattered by a concept class $\mathcal{H}$, that is, every finite path emanating from the root is consistent with a concept $h \in \mathcal{H}$. We say the subsequence $(X_{t_1}, \ldots, X_{t_k})$ is shattered by $\mathcal{H}$ as well.

The main idea of the algorithm is as follows. First, by looking at the history, the learner can detect a situation such that the length of the longest subsequences that are shattered by the partial concept class is $d$. Then we can use the algorithm to learn that partial concept class and ensure the algorithm only makes $O(\log T)$ mistakes.

We define the LCLL game as follows. The game has two players, $P_A$ and $P_B$. At each round $k$, each player can take the following actions:

- $P_A$ choose a sequence of instances of length $k$, $\mathbf{t}_k = (t_k^1, \ldots, t_k^k)$ such that $t_k^1 > t_{k-1}^{k-1}$ and $C_k = \{(y_{t_{k-1}^{k-1}+1,\varnothing}, \ldots, y_{t_k^1,\varnothing}^{u_1}, y_{t_k^1+1,u_1}, \ldots, y_{t_k^2,u_1}^{u_2}, \ldots, y_{t_k^k,u_{<k}}^{u_k}) : u \in \{0,1\}^k\}$, where $y_{k,u_{<i}}^0 \neq y_{k,u_{<i}}^i$ for all $i \leq k$. Here $u_{<i} = (u_1, \ldots, u_{i-1})$ and $u_{<1} = \varnothing$. Intuitively, at round $k$, $P_A$ proposes a level-constrained tree, with $(X_{t_k^1}, \ldots, X_{t_k^k})$ embedded in it.
- $P_B$ choose $g_{U_{k-1}}(\mathbf{t}_k, C_k) = (Y_{t_{k-1}^{k-1}}, \ldots, Y_{t_k^k}) \in C_k$. Intuitively, $P_B$ chooses a path from the level-constrained tree proposed by $P_A$.
- Update $U_k = U_{k-1} \cup \{(\mathbf{t}_k, C_k, (Y_{t_{k-1}^{k-1}}, \ldots, Y_{t_k^k}))\}$.
- $P_B$ wins the game in round $k$, if $\mathcal{H}_{U_k} = \emptyset$, where $\mathcal{H}_{U_k} = \{h \in \mathcal{H} : h(X_t) = Y_t, \forall t \leq t_k^k.\}$.

Following the tradition of game theory, we have the definition of *strategy* and *winning strategy*. A *strategy* is a way of playing that can be fully determined by the foregoing plays. A *winning strategy* is a strategy that necessarily causes the player to win no matter what action one's opponent takes.

The LCLL game is fully determined, as the membership of the winning set of $P_B$ is witnessed by a finite subsequence. Then we need the following lemma:

**Lemma 4.4.** *If $\mathcal{H}$ has no infinite indifferent LCLL tree, $P_B$*

---

[4]There are $2^{i-1}$ subtrees in total.

*has a winning strategy.*

To prove this lemma, we need to show that we can build an infinite indifferent LCLL tree based on the winning strategy of $P_A$. Then, due to the Borel Determinancy Theorem[5] and contrapositive, we know the lemma above is true. For brevity, we put the complete proof in Appendix B.

Notice that the winning strategy of $P_B$ is fully decided by $U$, thus, we can use $g_U$ to describe the winning strategy. Then we can write down Algorithm 1[6] and prove that for every sequence and a concept class with no infinite indifferent LCLL tree, it learns the target function with $O(\log T)$ mistakes.

---

**Algorithm 1** Learning algorithm for $\mathcal{H}$ with no infinite indifferent LCLL tree.

$k \leftarrow 1, U \leftarrow \{\}, t' \leftarrow 0$.
**for** $t = 1, 2, 3, \ldots$ **do**
  **if** $\exists t' \leq t_1 < t_2 < \cdots < t_k \leq t$ and $C_k = \{($
$y^{u_1}_{t^{k-1}_{k-1}+1,\varnothing}, \ldots, y^{u_1}_{t^1_k,\varnothing}, y_{t^1_k+1,u_1}, \ldots, y^{u_2}_{t^2_k,u_1} \cdots, y^{u_k}_{t^k_k,u_{<k}}) :$
$u \in \{0,1\}^k\}$ such that $g_U((t_1, \ldots, t_k), C_k) = (Y_{t'}, \ldots, Y_t)$ **then**
    Advance the game:
    $U \leftarrow U \cup \{((t_1, \ldots, t_k, C_k), g_U((t_1, \ldots, t_k), C_k))\}$.
    $k \leftarrow k + 1$.
    $L \leftarrow \emptyset$.
    $t' \leftarrow t$.
  **end if**
  Predict $\hat{Y}_t = \arg\max_y w\left(\mathcal{H}^{g_U}_{L \cup \{(X_t,y)\}}, X_{>t}\right)$.
  **if** $Y_t \neq \hat{Y}_t$ **then**
    $L \leftarrow L \cup \{(X_t, Y_t)\}$.
  **end if**
**end for**

---

To describe the weight function we use, we need several extra definitions. In the algorithm, we define the weight function as follows.

$$w(\mathcal{H}', X_{\geq t}) = \sum_{\substack{S : S \subseteq X_{\geq t} \\ \text{such that } S \text{ shattered by } \mathcal{H}'}} \frac{1}{n(S)^{d+1}}.$$

where $n(S)$ is the index of the last element in $S$, and $d$ largest length of the subsequences that is shattered by $\mathcal{H}'$. For example, if $S = \{X_{t_1}, \ldots, X_{t_k}\}$, then $n(S) = t_k$. Here, $\mathcal{H}^{g_U}$ is the partial concept class induced by $g_U$, which

---

[5]This is a classical result in game theory. We introduce those results in Appendix A.

[6]If the $\arg\max$ in the algorithm has multiple choices, it randomly outputs one of the choices. We use this tie-breaking rule for all $\arg\max$ in all of the algorithms.

---

is defined as follows.

$$\mathcal{H}^{g_{U_{k-1}}} =$$
$$\{h : \forall \mathbf{t}_k = (t^1_k, \cdots, t^k_k), t_{k-1} < t^1_k < \cdots < t^k_k,$$
$$\forall C_k, (h(X_{t^{k-1}_{k-1}}), \ldots, h(X_{t^k_k})) \neq g_U((t_1, \ldots, t_k), C_k)\},$$

where, for every $\mathbf{t}_k$, $C_k$ is defined as follows:

$$C_k = \{(y_{t^{k-1}_{k-1}+1,\varnothing}, \ldots, y^{u_1}_{t^1_k,\varnothing}, y_{t^1_k+1,u_1}, \ldots,$$
$$y^{u_2}_{t^2_k,u_1} \cdots, y^{u_k}_{t^k_k,u_{<k}}) : u \in \{0,1\}^k\}$$

Then we have the following lemma:

**Lemma 4.5.** *For any process $\{(X_i, Y_i)_{i \in \mathbb{N}}\} \in R(\mathcal{H})$, there exists $t_0$, such that for all $t \geq t_0$, algorithm 1 will not update $k$ and $U$ and for all $t < t_1 < t_2 < \cdots < t_k$, $(X_{t_1}, \ldots, X_{t_k})$ is not shattered by $\mathcal{H}^{g_U}$.*

This comes from the winning condition of $P_B$. We provide the complete proof in Appendix B.

According to Lemma 4.5, we know that if $\mathcal{H}$ does not have an indifferent LCLL tree, after $t_0$ rounds, the game will not advance. Therefore, for the rest of the sequence after $t_0$, the longest subsequence that is shattered by $\mathcal{H}^{g_U}$ has length $d$, then we want to use this fact to prove that this partial concept is transductively online learnable.

**Lemma 4.6.** *If the length of the longest subsequence that can be shattered by $\mathcal{H}$ is $d$, there is a transductive online learning algorithm that can learn $\mathcal{H}$ with $O(\log T)$ mistakes.*

The idea of this proof comes from the work of Alon et al. (2022). Briefly, we define a weight function and make sure that every time the algorithm makes a mistake, the weight function decreases by half. Then we provide an upper bound of the weight at the beginning, and a lower bound after $t$ rounds. Combining these two bounds, we have the number of mistakes made by the algorithm before round $t$. For brevity, the complete proof is provided in Appendix B.

**Lemma 4.7.** *If $\mathcal{H}$ has an infinite indifferent Littlestone tree, the adversary may force any learning algorithm to make $\Omega(\log T)$ mistakes.*

The idea of the proof is to run the random walk from the root of the infinite indifferent Littlestone tree and use the indifferent property to label all the nodes not visited by the walk. Therefore, the adversary may push the learner to make a mistake at each depth, and that pushes the learner to make $\Omega(\log T)$ mistakes. For brevity, the complete proof is provided in the appendix.

### 4.3. Constant Mistake Bound

In this subsection, we prove the constant mistake bound

**Lemma 4.8.** *If $\mathcal{H}$ does not have an infinite indifferent Littlestone tree, there is an algorithm $\mathcal{A}$ such that, for every $(\mathbb{X}, \mathbb{Y}) \in R(\mathcal{H})$, $M(\mathcal{A}, (\mathbb{X}, \mathbb{Y}), T) = O(1)$.*

To prove this lemma, we first define the following Littlestone game. In this game, there are two players, $P_A$ and $P_B$, and a fixed instance sequence $\mathbb{X}$. At each round $k$, each player can take the following actions:

- $P_A$ chooses two branches $B_k = \{(y_{t_{k-1}+1}, \ldots, y_{t_k-1}, y_{t_k}^0), (y_{t_{k-1}+1}, \ldots, y_{t_k-1}, y_{t_k}^1)\}$ and $t_k > t_{k-1}$.
- $P_B$ choose a branch $(Y_{t_{k-1}+1}, \ldots, Y_{t_k}) \in B_k$
- Update $U_k = U_{k-1} \cup \{(t_k, B_k, (Y_{t_{k-1}+1}, \ldots, Y_{t_k}))\}$.
- $P_B$ wins the game in round $k$, if $\mathcal{H}_{U_k} = \emptyset$. Here $\mathcal{H}_{U_k} = \{h \in \mathcal{H} : h(X_t) = Y_t, \forall t \le t_k\}$.

This game is fully determined, as the membership of the winning set of $P_B$ is witnessed by a finite subsequence. Then we need the following lemma.

**Lemma 4.9.** *If $\mathcal{H}$ has no infinite indifferent Littlestone tree, $P_B$ has a winning strategy.*

The proof is similar to the proof of Lemma 4.4, and the main tool is also the Borel Determinacy Theorem. For brevity, the complete proof is presented in the appendices.

Notice that $P_B$'s winning strategy is fully determined by $U$, so we use $g_U$ to stand for it. Then we can use $P_B$'s winning strategy to get Algorithm 2 that makes only finite mistakes for every realizable sequence $(\mathbb{X}, \mathbb{Y}) \in R(\mathcal{H})$. Here, $\mathcal{H}^{g_U}$ is the partial concept class induced by the winning strategy $g_U$, which is defined as follows.

$$\mathcal{H}^{g_U} = \{h : \forall t_k, \forall B_k, (h(X_{t_{k-1}}), \ldots, h(X_{t_k})) \ne g_U(t_k, B_k)\};$$

where, for every $t_k$, $B_k$ is defined as follows:

$$B_k = \{(y_{t_{k-1}+1}, \ldots, y_{t_k-1}, y_{t_k}^0), (y_{t_{k-1}+1}, \ldots, y_{t_k-1}, y_{t_k}^1)\}$$

---

**Algorithm 2** Learning algorithm for $\mathcal{H}$ with no infinite indifferent Littlestone tree

---

$U \leftarrow \{\}. \ t' \leftarrow 0.$
**for** $t = 1, 2, 3, \ldots$ **do**
    **if** $t \ge t'$ and there exists $B_k = \{(y_{t_{k-1}+1}, \ldots, y_{t_k-1}, y_{t_k}^0), (y_{t_{k-1}+1}, \ldots, y_{t_k-1}, y_{t_k}^1)\}$, such that $g_U(t, B_k) = (Y_{t'+1}, \ldots, Y_t)$ **then**
        Update the game:
        $U \leftarrow U \cup \{(t, B_k, (Y_{t'+1}, \ldots, Y_t))\}. \ t' \leftarrow t.$
    **end if**
    **if** $\exists h \in \mathcal{H}^{\hat{g}_U}, h(X_t) = y.$ **then**
        Predict $\hat{Y}_t = y.$
    **end if**
**end for**

---

**Lemma 4.10.** *For every sequence $(\mathbb{X}, \mathbb{Y}) \in R(\mathcal{H})$, if $\mathcal{H}$ does not have an infinite indifferent Littlestone tree, there exists a constant $t_0$, such that for every $t \ge t_0$, $X_t$ is not shattered by $\mathcal{H}^{g_U}$.*

*Proof.* By its definition, a winning strategy of $P_B$ leads to the winning condition of $P_B$. By the definition of $P_B$'s winning condition, we know that there is a constant $k$, such that for every $t_k > t_{k-1}$, $X_{t_k}$ is not shattered by $\mathcal{H}^{g_U}$. That finishes the proof. $\square$

According to Lemma 4.10 and Definition 4.3, we know that for all $t > t_0$, every $h \in \mathcal{H}^{g_U}$ satisfies $h(X_t) = Y_t$. Thus, the prediction is the true label $Y_t$. Therefore, the algorithm makes a finite number of mistakes.

## 5. Agnostic Setting

In this section, we present the high-level proof ideas of the learnability results in the agnostic setting, which is formally stated as Theorem 2.9. All the complete proofs are postponed to Appendix C for brevity.

To prove the upper bound, we need to first construct the experts based on the learning algorithm for the realizable case. An expert here is an algorithm with two hardcoded inputs, $I$ and $J$. $I = (X_{\le k}, Y_{\le k}^*)$ for a constant $k \in \mathbb{N}$, which is a hallucinated sequence for the game updating part, and $J \subseteq \mathbb{N}$ marks the indices of the mistakes made by the realizable algorithm when $Y_t = \mathcal{Y}_t^*$ during learning $(\mathbb{X}, \mathbb{Y}^*)$ after the game updating part. When the time is in the set of $I$, the prediction of the expert is given by the hallucinated label stored by $I$. For the rest, the expert will predict a random label if the index of the round is in the set $J$. We can prove the following lemma for the experts defined above.

**Lemma 5.1.** *If $\mathcal{H}$ has no infinite indifferent LCLL tree, for every realizable sequence $(\mathbb{X}, \mathbb{Y}) \in R(\mathcal{H})$, we have a sequence $\{j_T\}_{T \in \mathbb{N}}$ satisfies $\log j_T = O((\log T)^2)$, such that for every large enough time $T$, we have an expert $e_{i,j}$ with $j \le j_T$, such that for all $t \le T$, $Y_t = e_{i,j}(X_t)$ except for at most $O(\log T)$ times.*

Then, we can use the *Squint* algorithm from the work of Koolen & van Erven (2015) with non-uniform initial weights. For each expert $e_{i,j}$, we set its initial weight as $\pi_{i,j} = \frac{1}{i(i+1)j(j+1)}$ and this forms a distribution, as $\pi_{i,j} = \frac{1}{i(i+1)j} - \frac{1}{i(i+1)(j+1)}$ and $\sum_{j=1}^{\infty} p_{i,j} = \frac{1}{i} - \frac{1}{i+1}$, the sum of $\pi_{i,j}$ reaches 1 when $i$ and $j$ goes to infinity. According to Theorem 3 in the work of Koolen & van Erven

(2015), we have the following upper bound for the regret

$$\sum_{t=1}^{T} \mathbb{I}\left[\hat{Y}_t \neq Y_t\right] - \sum_{t=1}^{T} \mathbb{I}\left[e_{i,j}(X_t) \neq Y_t\right]$$
$$\leq O\left(\sqrt{V_{i,j} \log \frac{\log V_{i,j}}{\pi_{i,j}}} + \log \frac{1}{\pi_{i,j}}\right).$$

Here, the $V_{i,j}$ is the sum of the squares of the difference between the algorithm's mistake and the expert $e_{i,j}$'s mistake in each round. In other words, we have

$$V_{i,j} = \sum_{t=1}^{T} \left(\mathbb{I}\left[\hat{Y}_t \neq Y_t\right] - \mathbb{I}\left[e_{i,j}(X_t) \neq Y_t\right]\right)^2.$$

Notice that $\left(\mathbb{I}\left[\hat{Y}_t \neq Y_t\right] - \mathbb{I}\left[e_{i,j}(X_t) \neq Y_t\right]\right)^2$ is either 1 or 0, we have $V_{i,j} \leq T$. The regret of this algorithm is upper bounded by:

$$O\left(\sqrt{V_{i,j} \log \frac{\log V_{i,j}}{\pi_{i,j}}} + \log \frac{1}{\pi_{i,j}}\right)$$
$$= O\left(\sqrt{T \log \log T + T(\log i + \log j)} + (\log i + \log j)\right).$$

On the other hand, we also prove a lower bound in the agnostic setting, that is, no learning algorithm for a concept class that includes more than two concepts can ensure a $o(\sqrt{T})$ number of mistakes for every sequence $(\mathbb{X}, \mathbb{Y}, \mathbb{Y}^*)$. Formally,

**Lemma 5.2.** *For every concept class $\mathcal{H}$ containing two concepts $h_1, h_2$ and we have $x, h_1(x) \neq h_2(x)$, for every learning algorithm $\mathcal{A}$, there is a sequence $(\mathbb{X}, \mathbb{Y}, \mathbb{Y}^*)$ such that $Regret(\mathcal{A}, (\mathbb{X}, \mathbb{Y}, \mathbb{Y}^*), T) \neq o(\sqrt{T})$.*

To prove this lemma, we need to construct an infinite sequence such that no online learning algorithm can transductively learn it with regret of $o(\sqrt{T})$. The sequence $\mathbb{X}$ is an infinite sequence of $X_t = x$, such that $h_1(x) \neq h_2(x)$. Then we use some probabilistic statements, that is, if we independently uniformly randomly pick $h_1(x)$ or $h_2(x)$ as $Y_t$, there exists an infinite sequence $\mathbb{Y}$ and $\mathbb{Y}^* = \{h_1(X_i)\}_{i \in \mathbb{N}}$ or $\{h_2(X_i)\}_{i \in \mathbb{N}}$ such that the regret of any online learning algorithm is not $o(\sqrt{T})$ with probability more than 0. To prove this, we divide the infinite sequence into blocks of increasing size and use Khinchine's Inequality on each block to show the expected regret on that block is $\Omega(\sqrt{T})$. Then, by Azuma's inequality, we can bound the probability of generating the sequence that pushes any algorithm to suffer a $\Omega(\sqrt{T})$ regret. Then we can extend this result on blocks to the result on the infinite sequence by the reversed Fatou's lemma. For brevity, the complete proof is provided in the Appendix C.

## 6. Conclusion and Future Directions

In this paper, we investigate universal multiclass transductive online learning. We prove a trichotomy for the realizable setting. To describe the trichotomy, we define a new combinatorial structure, the Level-Constrained Littlestone-Littlestone tree, and emphasize the indifferent property of trees. We then provide the $\tilde{O}(\sqrt{T})$ regret learning algorithm for the agnostic case when $\mathcal{H}$ has no infinite indifferent LCLL tree. We also extend the learnability result to the case of learners with knowledge only of a stochastic process from which the instance sequence is sampled, describing the condition in which all processes admit universal multiclass online learning.

Finally, we want to provide several interesting future directions:

- First, for the agnostic case, there is a poly-logarithmic gap between the upper and lower bounds of the regret. Is it possible to tighten this gap?
- Second, is it possible to characterize the universal transductive online learnability in other settings? For example, multiclass classification with bandit feedback, real-valued function regression, etc. As we have shown, this setting is closely related to the condition where all processes admit universal online learning; this question itself is also very interesting.

## Acknowledgements

We thank the reviewers who provided useful suggestions on improving the quality of this paper. **SH** acknowledges support by grant no. 2024243 from the United States - Israel Binational Science Foundations (BSF).

## Impact Statement

This paper presents work whose goal is to advance the field of Learning Theory. There are many potential societal consequences of our work, none of which we feel must be specifically highlighted here.

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

# A. Gale-Stewart Game

In this section, we very briefly review the basic notions from the classical theory of infinite games.

In a two-player game, there are two players, $A$ and $B$. The players have an action space $\mathcal{A}$. At round $t$, the player $A$ chooses an action from the action space, and then the player $B$ also chooses an action from the action space. Thus, each play of the game can be expressed as a sequence of actions, $g = (a_1, a_2, a_3, a_4, \ldots)$, where $a_i \in \mathcal{A}$ for every $i$. The set of all such sequences is denoted as $\mathcal{A}^\omega$. Here $\mathcal{A}$ is equipped with the discrete topology and $\mathcal{A}^\omega$ is equipped with the generated product topology. A fixed set determines the result of the game, known as the winning set, $W$. If the play of the game $g \in W$, the player $A$ wins, and player $B$ wins otherwise.

A **strategy** is a function that generates the next action for the player based on the states of the game (which can be thought of as the history of actions). If the strategy of player $A$ leads to the winning of player $A$ regardless of the actions of player $B$ is called the **winning strategy** of player $A$. If one player or the other has a winning strategy and no other cases, the game is **determined**. Then we have the formal theorem about the determinacy of the game.

**Theorem A.1** (Gale & Stewart (1953)). *If $W$ is an open set, then the game is determined.*

The proof is short and intuitive. We would like to point the readers to the original paper for the proof. This theorem is called the Borel Determinacy Theorem, because Borel sets are the smallest $\sigma$-algebra of subsets of $\mathcal{A}^\omega$ that contains all open sets.

# B. Omitted Proofs for the Realizable Setting

### B.1. Proof of Lemma 4.1

*Proof.* We construct a sequence based on the properties of an infinite indifferent LCLL tree, so any learning algorithm cannot make sublinear expected mistakes.

First, we start from an infinite indifferent LCLL tree and then modify it such that the length of the sequence in each node increases exponentially. In other words, we hope the $k$-th node in the Breadth-First-Search (BFS) order contains a $2^{k-1}$ long sequence. We may reach this target by recursively modifying the chosen indifferent LCLL tree. Starting from the root of the tree, for each node that does not satisfy our requirement, we promote one of its descendants to replace that node, such that the number of elements in that node is large enough.

Then the data sequence $\mathbb{X}$ we look at is the following: For the modified indifferent LCLL tree, starting from the root with $\mathbb{X} = \emptyset$, traverse the tree in Breadth-First-Order and every time reach a new node, append the sequence in that node to $\mathbb{X}$.

Next, we choose the true label $Y_t$ for each $X_t$ and make $(\mathbb{X}, \mathbb{Y})$ a realizable sequence. First, we take a random walk on the modified indifferent LCLL tree. For every $X_t$ that is in a node visited by that random walk (in-branch node), we decide $Y_t$ by the label of the edge visited and the function $\mathbf{y}_n$. For every $X_t$ that is not in a node visited by that random walk (out-branch node), we will find the closest in-branch node after this out-branch node and label $X_t$ by the function agreed on by all descendants of that in-branch node. As the tree is indifferent, we can find such a function.

Then we prove this is a sequence that pushes any transductive online learning algorithm to make more than $o(T)$ mistakes in expectation. Consider the node in depth $d$ visited by the random walk, which is the $K_d$-th node in the BFS order and labeled with a $2^{K_d-1}$ length sequence. Due to the property of the indifferent LCLL tree, we know that each $X_t$ in the $K_d$-th node can take any label sequence on the edge adjacent to that node. Thus, the best algorithm still makes $2^{K_d-2}$ mistakes in expectation. Due to the definition of the sequence, we know the length of the sequence after the $K_d$-th node is $n_{K_d} = \sum_{i=1}^{K_d} 2^{i-1} = 2^{K_d} - 1$. Thus, we have the following inequality for every $d$,

$$\mathbb{E}\left[\frac{1}{n_{K_d}} \sum_{t=1}^{n_{K_d}} \mathbb{I}\left[\hat{h}_{t-1}(X_t) \neq Y_t\right] \middle| K_d\right] \geq \frac{1}{4}. \tag{1}$$

Thus, we have

$$\limsup_{d \to \infty} \mathbb{E}\left[\frac{1}{n_{K_d}} \sum_{t=1}^{n_{K_d}} \mathbb{I}\left[\hat{h}_{t-1}(X_t) \neq Y_t\right] \middle| K_d\right] \geq \frac{1}{4}.$$

Then consider the expected number of mistakes, which is

$$\mathbb{E}\left[\limsup_{n\to\infty}\frac{1}{n}\sum_{t=1}^{n}\mathbb{I}\left[\hat{h}_{t-1}(X_t)\neq Y_t\right]\right]$$

$$\geq\mathbb{E}\left[\limsup_{d\to\infty}\frac{1}{n_{K_d}}\sum_{t=1}^{n_{K_d}}\mathbb{I}\left[\hat{h}_{t-1}(X_t)\neq Y_t\right]\right]$$

$$=\mathbb{E}\left[\mathbb{E}\left[\limsup_{d\to\infty}\frac{1}{n_{K_d}}\sum_{t=1}^{n_{K_d}}\mathbb{I}\left[\hat{h}_{t-1}(X_t)\neq Y_t\right]\Bigg|K_d\right]\right]$$

$$\geq\mathbb{E}\left[\limsup_{d\to\infty}\mathbb{E}\left[\frac{1}{n_{K_d}}\sum_{t=1}^{n_{K_d}}\mathbb{I}\left[\hat{h}_{t-1}(X_t)\neq Y_t\right]\Bigg|K_d\right]\right]$$

$$\geq\frac{1}{4}.$$

For the sequence of the average number of mistakes, we take the sub-sequence that only contains the $K_d$-th element, then the limit superior of the sub-sequence is smaller than the limit superior of the whole sequence. That is the second line in the equations above. The third line is due to the law of total expectation. Then, as the average number of mistakes is always positive and bounded by 1, we can use the reversed Fatou's lemma to get the inequality in the fourth line.

Therefore, there exists a realizable sequence such that any realizable online learner will make more than a sublinearly expected number of mistakes. □

### B.2. Proof of $O(\log T)$ Upper Bound

In this section, we provide the complete proof of the $O(\log T)$ upper bound, when $\mathcal{H}$ does not have an infinite indifferent LCLL tree.

*Proof of Lemma 4.4.* Due to the Borel Determinacy Theorem, we know that if $P_A$ does not have a winning strategy, $P_B$ has a winning strategy. Thus, we only need to build an infinite indifferent LCLL tree based on the fact that $P_A$ has a winning strategy. This is a constructive proof. We show how to recursively build an infinite indifferent LCLL tree.

We build the infinite indifferent LCLL tree in the Breadth-First-Search (BFS) order from the root. We first take the depth-1 LCL tree proposed by $P_A$ and use it as the root and its two edges.

Then for the $i$-th node $v_i$, suppose the $(i-1)$-th node at depth $k$ is labeled by $(X_{j_1},\ldots,X_{j_k})$. We also suppose $v_i$'s parent node is labeled by $(X_{\ell_1},\ldots,X_{\ell_{k-1}})$. Because $P_A$ has a winning strategy, no matter which branch is chosen by $P_B$, for any $n\in\mathbb{N}$, $P_A$ can propose a sequence $(X_{i_1},\ldots,X_{i_n})$ shattered by $\mathcal{H}_{(X_{\leq i_1},Y_{\leq i_1})}$. This comes from $P_A$'s action and Definition 4.3. Here, the labels $Y_i$ for $\ell_{k-1}\leq i\leq i_1$ are proposed by $P_A$ but make sure it is realizable by $\mathcal{H}$, as otherwise $P_B$ wins the game.

Then we can take the first $k$ instances in the sequence $(X_{i_1},\ldots,X_{i_n})$ and use it to label $v_i$. By the definition of shattering, we know that we can build a depth-$k$ LCL tree with label sequences $\{(y_{i_1,\varnothing}^{u_1},\ldots,y_{i_k,u_{<k}}^{u_k}):u\in\{0,1\}^k\}$, where $u_{<k}=(u_1,u_2,\ldots,u_{k-1})$. Thus, we can use these label sequences to label the edges connecting $v_i$ and its children.

Notice that the sequence $(X_{i_1},\ldots,X_{i_k})$ is shattered by $\mathcal{H}_{(X_{\leq i_1},Y_{\leq i_1})}$, all functions of its descendants agree on all the instances before $i_1$. All nodes before the $i$-th node in BFS order are labeled by instances before $X_{i_1}$. Thus, all the functions of the descendants of $v_i$ agree on all nodes before. Therefore, $P_A$'s winning strategy leads to an infinite indifferent LCLL tree for $\mathcal{H}$. □

*Proof of Lemma 4.5.* By the definition of the winning strategy, it leads to a winning condition for the player $P_B$. By the definition of $P_B$'s winning condition, we know that there exists a $k$ such that $\mathcal{H}_{\mathbf{t}_1,C_1,g_1,\ldots,\mathbf{t}_k,C_k,g_k}=\emptyset$, which means for all $t<t_1<t_2<\cdots<t_k$, the sequence $(X_{t_1},\ldots,X_{t_k})$ is not shattered by $\mathcal{H}^{g_{U_{k-1}}}$. That finishes the proof. □

We then prove that if the length of the longest subsequences shattered by $\mathcal{H}'$ is $d$, there is a learning algorithm that can learn it with $O(\log T)$ mistakes. For completeness, we provide the subroutine as an algorithm and restate the lemma here.

---

**Algorithm 3** Subroutine for learning at most length-$d$ subsequences are shattered by $\mathcal{H}'$.

---

$L \leftarrow \emptyset$.
**for** $t = 1, 2, 3, \ldots$ **do**
    Predict $\hat{Y}_t = \arg\max_y w(\mathcal{H}'_{L \cup \{(X_t, y)\}}, X_{\geq t})$.
    **if** $Y_t \neq \hat{Y}_t$ **then**
        $L \leftarrow L \cup \{(X_t, Y_t)\}$.
    **end if**
**end for**

---

In the algorithm 3, we define the weight function as follows.

$$w(\mathcal{H}', X_{\geq t}) = \sum_{\substack{S: S \subseteq X_{\geq T} \\ \text{such that } S \text{ shattered by } \mathcal{H}'}} \frac{1}{n(S)^{d+1}}, \tag{2}$$

where $n(S)$ is the index of the last element in $S$, $d$ is the length of the longest subsequences that are shattered by the partial concept class $\mathcal{H}'$, and $X_{>t} = \{X_i\}_{i>t}$. For example, if $S = \{X_{t_1}, \ldots, X_{t_k}\}$, then $n(S) = t_k$. Then we can restate and prove Lemma 4.6 here.

**Lemma B.1** (Restate of Lemma 4.6). *If the length of the longest subsequences that are shattered by the partial concept class $\mathcal{H}'$ is d, Algorithm 3 can learn it with $O(\log T)$ mistakes.*

*Proof.* Notice that if $S$ is shattered by $\mathcal{H}_{\{(X_i, Y_i)\}_{i<t} \cup \{(X_t, y_t^0)\}}$ and $\mathcal{H}_{\{(X_i, y_i)\}_{i<t} \cup \{(X_t, y_t^1)\}}$, $S \cup \{X_t\}$ is shattered by $\mathcal{H}_{\{(X_i, Y_i)\}_{i<t}}$. Thus, if the subset $S$ contributes twice on the right-hand side of the inequality, it also contributes at least twice on the left-hand side. (One comes from $S$, the other comes from $S \cup \{X_t\}$.) Thus, we have the following inequality for $\mathbb{X}$,

$$w(\mathcal{H}_{(X_{<t}, Y_{<t})}, X_{\geq t}) \geq w(\mathcal{H}_{(X_{<t}, Y_{<t}) \cup \{(X_t, y_t^0)\}}, X_{>t}) + w(\mathcal{H}_{(X_{<t}, Y_{<t}) \cup \{(X_t, y_t^1)\}}, X_{>t}),$$

where $(X_{<t}, Y_{<t}) = \{(X_i, Y_i)\}_{i<t}$. Every time when the algorithm makes a mistake, we have $w(\mathcal{H}_{(X_{<t}, Y_{<t}) \cup \{(X_t, \hat{Y}_t)\}}, X_{>t}) \geq w(\mathcal{H}_{(X_{<t}, Y_{<t}) \cup \{(X_t, Y_t)\}}, X_{>t})$. Thus, we have

$$w(\mathcal{H}_{(X_{<t}, Y_{<t}) \cup \{(X_t, Y_t)\}}, X_{>t}) \leq \frac{1}{2} w(\mathcal{H}_{(X_{<t}, Y_{<t})}, X_{\geq t})$$

Then we need to get an upper bound of $w(\mathcal{H}', \mathbb{X})$, notice that for the entire sequence, we are taking the sum over all possible $n$, then we need to bound the number of the sub-sequences ending at $X_n$ and can be shattered by $\mathcal{H}'$. We know that it is the number of subsets of $\{1, 2, \ldots, n\}$, who contains $n$ and has less than or equal to $d$ elements, that is

$$\sum_{i=1}^{d-1} \binom{n-1}{i} \leq n^{d-1}.$$

Thus, we have

$$w(\mathcal{H}', \mathbb{X}) \leq \sum_{n=1}^{\infty} \frac{n^{d-1}}{n^{d+1}} = \sum_{n=1}^{\infty} \frac{1}{n^2} \leq \frac{\pi^2}{6}.$$

Then we provide the lower bound of the weight function at each round $T$. Consider the last mistake the algorithm made before round $T$, before that mistake, we know at least a singleton $\{x_{t'}\}$ is shattered by the concept class. Thus, the weight function at round $T$ is greater than $\frac{1}{T^{d+1}}$. Assume the algorithm makes $m(T)$ number of mistakes before round $T$. We have

$$\frac{\pi^2}{6} \geq w(\mathcal{H}', \mathbb{X}) \geq 2^{m(T)-1} \frac{1}{T^{d+1}}.$$

Therefore, we have $u(T) \leq (d+1) \log T + 1$, which is $O(\log T)$. $\qquad\square$

**B.3. Proof of $\Omega(\log T)$ lower bound**

*Proof of Lemma 4.7.* We show the lower bound by using the property of an infinite indifferent Littlestone tree. The instance sequence we take is the sequence of all instances that label the node. We list them in the BFS order.

Then we choose the true label $Y_t$ for each instance $X_t$ as follows. We take a random walk starting from the root of the tree. For each node visited by that random walk (in-branch node), let $Y_t$ be the label of the edge visited after that node. For those nodes that are not visited, we take the in-branch node that comes after it, and label the node by the function agreed on by all the descendants of that in-branch node. We can always find that function due to the indifferent property.

Then we can prove this sequence pushes any transductive online learning algorithm to make $\Omega(\log T)$ mistakes. Notice that for this Littlestone tree, every level, there is a node (visited by the random walk) such that any algorithm will make a mistake with probability $\frac{1}{2}$. Notice that for every $T$, there are at least $\lfloor \log T \rfloor$ levels. Thus, there are $\lfloor \log T \rfloor$ nodes where the algorithm will make a mistake with probability half. Therefore, the number of mistake in expectation is $\frac{\lfloor \log T \rfloor}{2} \geq \frac{\log T}{2} - 1$, which is $\Omega(\log T)$. □

**B.4. Proof of Constant Upper Bound**

*Proof of Lemma 4.9.* Due to the Borel Determinacy Theorem, we know that if $P_A$ does not have a winning strategy, $P_B$ has a winning strategy. Thus, we only need to show how to build an infinite indifferent Littlestone tree from the winning strategy of $P_A$. We can recursively build an infinite indifferent Littlestone tree in the Breadth-First-Search (BFS) order from the winning strategy of $P_A$.

Starting from the root, we can label the root and its two edges by the instance $X_{t_1}$ and two labels $\{y_{t_1}^0, y_{t_1}^1\}$ proposed by $P_A$. Then for the $i$-th node $v_i$ at depth $k$, assume the $(i-1)$-th node is labeled by $X_{t_{i-1}}$, and $v_i$'s parent is labeled by $X_{t_{k-1}}$.

Because $P_A$ has a winning strategy, no matter which branch $P_B$ takes, $P_A$ can propose a $t_k > t_{i-1}$ and two branches $\{(y_{t_{k-1}+1}, \ldots, y_{t_k-1}, y_{t_k}^0), (y_{t_{k-1}+1}, \ldots, y_{t_k-1}, y_k^1)\}$, such that $X_{t_k}$ is shattered by $\mathcal{H}_{(X_{<t_k}, Y_{<t_k})}$. Here the sequence $((X_{t_{k-1}+1}, Y_{t_{k-1}+1}), \ldots, (X_{t_k-1}, Y_{t_k-1}))$ is chosen by $P_A$, but need to ensure that it is realizable. Thus, we use $X_{t_k}$ to label $v_i$ and $\{y_{t_k}^0, y_{t_k^1}\}$ to label its two edges connecting its children.

We know all the nodes come before $v_i$ in BFS order is labeled by an instance before $X_{t_{i-1}}$ Thus, all the functions of the descendants of $X_{t_k}$ are in the concept class $\mathcal{H}_{(X_{<t_k}, Y_{<t_k})}$, where $t_{i-1} < t_k$. Thus, they agree on all nodes before $v_i$ in BFS order. Therefore, the winning strategy of $P_A$ leads to an infinite indifferent Littlestone tree. □

# C. Agnostic Setting

In this part, we show that whether $\mathcal{H}$ has an infinite indifferent LCLL tree characterizes the universal transductive online learnability in the agnostic setting as well. We use the learning with experts algorithm to handle this problem. This is a traditional way to expand the learning algorithm for the realizable setting to a learning algorithm for the agnostic setting. Specifically, we use the algorithm called *Squint* from the work of Koolen & van Erven (2015) with non-uniform initial weights. As we mentioned in Section 5, for each expert $e_{i,j}$, we set its initial weight as $\pi_{i,j} = \frac{1}{i(i+1)j(j+1)}$.

And this provides the upper bound of the regret of this algorithm as

$$\sum_{t=1}^{T} \mathbb{I}\left[\hat{Y}_t \neq Y_t\right] - \sum_{t=1}^{T} \mathbb{I}\left[e_{i,j}(X_t) \neq Y_t\right]$$

$$\leq O\left(\sqrt{V_{i,j} \log \frac{\log V_{i,j}}{\pi_{i,j}}} + \log \frac{1}{\pi_{i,j}}\right) = O\left(\sqrt{T \log \log T + T(\log i + \log j)} + (\log i + \log j)\right). \tag{3}$$

Then we only need to build the experts and show that for any realizable sequence $(\mathbb{X}, \mathbb{Y}^*) \in R(\mathcal{H})$, there is an expert whose prediction at time $t$, when $Y_t^* = Y_t$, is different from $Y_t^*$ for at most $\log T$ different time $t$, for all $t \leq T$.

An expert here is an algorithm with two hardcoded inputs, $I$ and $J$. $I = (X_{\leq k}, Y_{\leq k}^*)$ for a constant $k \in \mathbb{N}$, which is a hallucinated sequence for the game updating part, and $J \subseteq \mathbb{N}$ marks the indices of the mistakes made by the realizable algorithm when $Y_t = \mathcal{Y}_t^*$ during learning $(\mathbb{X}, \mathbb{Y}^*)$ after the game updating part. The expert is defined as follows. In Algorithm 4, $Y_t^*$ is the hallucinated label stored by $I$ and $Y_t$ is the true label given by the adversary. Then we need the

---

**Algorithm 4** Expert $I, J$ (indexed by $e_{i,j}$).

---

$U \leftarrow \{\}. \ t' \leftarrow 0. k \leftarrow 1. L \leftarrow \{\}.$
**for** $t = 1, 2, 3, \ldots$ **do**
  **if** $t \leq k$ **then**
    **if** $\exists t' \leq t_1 < t_2 < \cdots < t_k \leq t$ and $C_k = \{(y_{t_{k-1}^{k-1}+1, \varnothing}, \ldots, y_{t_k^1, \varnothing}^{u_1}, y_{t_k^1+1, u_1}, \ldots, y_{t_k^2, u_1}^{u_2} \ldots, y_{t_k^k, u_{<k}}^{u_k}) : u \in$
    $\{0,1\}^k\}$ such that $g_U((t_1, \ldots, t_k), C_k) = (Y_{t'}, \ldots, Y_t)$ **then**
      Advance the game:
      $U \leftarrow U \cup \{((t_1, \ldots, t_k, C_k), g_U((t_1, \ldots, t_k), C_k))\}.$
      $k \leftarrow k + 1.$
      $t' \leftarrow t.$
    **end if**
    Predict $\hat{Y}_t = Y_t^*.$
  **else**
    Predict $\hat{Y}_t = \arg\max_y w(\mathcal{H}_{L \cup \{(X_t, y)\}}^{g_U}, X_{\geq t}).$
    **if** $t \in J$ **then**
      $L \leftarrow L \cup \{(X_t, Y_t)\}.$
    **end if**
  **end if**
**end for**

---

following lemma about the expert.

**Lemma C.1.** *If $\mathcal{H}$ has no infinite indifferent LCLL tree, for every realizable sequence $(\mathbb{X}, \mathbb{Y}) \in R(\mathcal{H})$, we have a sequence $\{j_T\}_{T \in \mathbb{N}}$ satisfies $\log j_T = O((\log T)^2)$, such that for every large enough time $T$, we have an expert $e_{i,j}$ with $j \leq j_T$, such that for all $t \leq T$, $Y_t = e_{i,j}(X_t)$ except for at most $O(\log T)$ times.*

*Proof.* Firstly, because $\mathcal{Y}$ is countable and $k$ is finite, the number of different $I$'s is countable. Referring to Lemma 4.5, we know that for every realizable sequence $(\mathbb{X}, \mathbb{Y}) \in R(\mathcal{H})$, the game only updates finitely many times. Thus, there is an $I = (X_{\leq k}, Y_{\leq k})$, such that after $k$, the game does not update anymore, and we have all experts $e_{i,*}$ make no mistake during the game updating period.

Then we look at the second stage. For every large enough $T$, due to Lemma 4.6, there is an algorithm that only makes $O(\log T)$ number of mistakes. Thus, we have $j_T = T^{O(\log T)}$, which is the number of all possible subsets whose size is $O(\log T)$. So, there is a $j < j_T$ such that the input $J$ contains all the indices of the mistakes made by Algorithm 3 when learning $(\mathbb{X}, \mathbb{Y})$ after the game updating. Therefore, for every $t \leq T$, we have $Y_t = e_{i,j}(X_t)$ if $t \notin J$. That finishes the proof. $\square$

Then we need to extend this result to the agnostic case. By the definition of regret, we have

$$\text{Regret}(\mathcal{A}, (\mathbb{X}, \mathbb{Y}, \mathbb{Y}^*), T) = \mathbb{E}\left[\sum_{t=1}^{T} \left(\mathbb{I}\left[Y_t \neq \hat{Y}_t\right] - \mathbb{I}\left[Y_t \neq Y_t^*\right]\right)\right]$$

$$= \mathbb{E}\left[\sum_{t=1}^{T} \left(\mathbb{I}\left[Y_t \neq \hat{Y}_t\right] - \mathbb{I}\left[Y_t \neq e_{i,j}(X_t)\right] + \mathbb{I}\left[Y_t \neq e_{i,j}(X_t)\right] - \mathbb{I}\left[Y_t \neq Y_t^*\right]\right)\right]$$

$$= \mathbb{E}\left[\sum_{t=1}^{T} \left(\mathbb{I}\left[Y_t \neq \hat{Y}_t\right] - \mathbb{I}\left[Y_t \neq e_{i,j}(X_t)\right]\right)\right] + \mathbb{E}\left[\sum_{t=1}^{T} \left(\mathbb{I}\left[Y_t \neq e_{i,j}(X_t)\right] - \mathbb{I}\left[Y_t \neq Y_t^*\right]\right)\right].$$

Notice that the first part is bounded by inequality 3, we only need to bound the second part. For a sequence $(\mathbb{X}, \mathbb{Y}, \mathbb{Y}^*)$, we can take the subsequence $(\mathbb{X}', \mathbb{Y}', \mathbb{Y}'^*) = \{(X_t, Y_t, Y_t^*) : Y_t = Y_t^*\}$. We can take $T' \leq T$, such that $\mathbb{E}[\sum_{t=1}^{T}(\mathbb{I}[Y_t \neq e_{i,j}(X_t)] - \mathbb{I}[Y_t \neq Y_t^*])] = \mathbb{E}[\sum_{t=1}^{T'}\mathbb{I}[Y_t' \neq e_{i,j}(X_t')]]$. By Lemma C.1, we know there is an expert $e_{i,j}$ for $j < j_{T'} < j_T$ such that $\sum_{t=1}^{T'}\mathbb{I}[Y_t' \neq e_{i,j}(X_t')] = O(\log T')$. Therefore, we have

$\mathbb{E}[\sum_{t=1}^{T} (\mathbb{I}[Y_t \neq e_{i,j}(X_t)] - \mathbb{I}[Y_t \neq Y_t^*])] = O(\log T)$. Combining with the bound for the first part, we have

$$\text{Regret}(\mathcal{A}, (\mathbb{X}, \mathbb{Y}, \mathbb{Y}^*), T) = O\left(\sqrt{T \log \log T + T(\log i + \log j) + (\log i + \log j)}\right) + O(\log T)$$

$$= O\left(\sqrt{T \log \log T + T \log j_T + \log j_T}\right)$$

$$= O(\sqrt{T} \log T).$$

Then we show the following lower bound.

**Lemma C.2.** *For every concept class $\mathcal{H}$ containing two concepts $h_1, h_2$ and we have $x$, $h_1(x) \neq h_2(x)$, for every learning algorithm $\mathcal{A}$, there is a sequence $(\mathbb{X}, \mathbb{Y}, \mathbb{Y}^*)$ such that $\text{Regret}(\mathcal{A}, (\mathbb{X}, \mathbb{Y}, \mathbb{Y}^*), T) \neq o(\sqrt{T})$.*

To prove this lemma, we need the following lemma.

**Lemma C.3** (Khinchine's Inequality; Lemma 8.2 in (Cesa-Bianchi & Lugosi, 2006)). *Let $T \in \mathbb{N}$, Let $\sigma_1, \ldots, \sigma_T$ be random variables sampled independently and uniform randomly from $\{\pm 1\}$. We have the following inequality:*

$$\mathbb{E}\left[\left|\sum_{t=1}^{T} \sigma_t\right|\right] \geq \sqrt{\frac{T}{2}}.$$

*Proof of Lemma C.2.* Consider the instance sequence $\mathbb{X} = \{X_t\}_{t \in \mathbb{N}}$, such that for all $t$, $X_t = x$. In the true label sequence $\mathbb{Y} = \{Y_t\}_{t \in \mathbb{N}}$, for every $t$, $Y_t$ takes $h_1(x)$ or $h_2(x)$ independently uniformly randomly. As for $\mathbb{Y}^* = \{Y_t^*\}_{t \in \mathbb{N}}$, $Y_t^* = h_1(x)$ for every $t$, or $Y_t^* = h_2(x)$ for every $t$.

Then consider the following random variable:

$$R_i = \mathbb{E}\left[\sum_{t=T_{i-1}+1}^{T_i} \mathbb{I}\left[Y_t \neq \hat{Y}_t\right] - \min_{h \in \{h_1, h_2\}} \left(\sum_{t=T_{i-1}+1}^{T_i} \mathbb{I}[Y_t \neq h(X_t)]\right) \Bigg| \mathbb{Y}\right],$$

where $T_i = \sum_{j=1}^{i} 2^{4^j}$. And we have $T_i - T_{i-1} = 2^{4^i}$. Then, we consider the expectation of $R_i$,

$$\mathbb{E}[R_i] = \mathbb{E}\left[\mathbb{E}\left[\sum_{t=T_{i-1}+1}^{T_i} \mathbb{I}\left[Y_t \neq \hat{Y}_t\right] - \min_{h \in \{h_1, h_2\}} \left(\sum_{t=T_{i-1}+1}^{T_i} \mathbb{I}[Y_t \neq h(X_t)]\right) \Bigg| \mathbb{Y}\right]\right]$$

$$= \mathbb{E}\left[\mathbb{E}\left[\sum_{t=T_{i-1}+1}^{T_i} \mathbb{I}\left[Y_t \neq \hat{Y}_t\right] \Bigg| \mathbb{Y}\right]\right] - \mathbb{E}\left[\min_{h \in \{h_1, h_2\}} \left(\sum_{t=T_{i-1}+1}^{T_i} \mathbb{I}[Y_t \neq h(X_t)]\right)\right]$$

$$= \frac{2^{4^i}}{2} - \mathbb{E}\left[\min\{r_i, 2^{4^i} - r_i\}\right],$$

where $r_i = \sum_{t=T_{i-1}+1}^{T_i} \mathbb{I}[Y_t \neq h_2(X_t)]$. This comes from the linearity of expectation and the fact that the algorithm is independent of the true label sequence. Therefore,

$$\mathbb{E}[R_i] = \mathbb{E}\left[\left|\frac{2^{4^i}}{2} - r_i\right|\right] = \mathbb{E}\left[\left|\sum_{\ell=1}^{2^{4^i}} \frac{1}{2} - \left(\frac{1}{2} + \frac{\sigma_i^\ell}{2}\right)\right|\right] = \frac{1}{2}\mathbb{E}\left[\left|\sum_{\ell=1}^{2^{4^i}} \sigma_i^\ell\right|\right],$$

where $\sigma_i^\ell = 1$ if $\mathbb{I}[Y_t \neq h_2(X_t)]$ for $t = T_{i-1} + \ell$, and $\sigma_i^\ell = -1$ if $\mathbb{I}[Y_t \neq h_1(X_t)]$ for $t = T_{i-1} + \ell$. Due to Lemma C.3, we have

$$\mathbb{E}\left[\left|\sum_{\ell=1}^{2^{4^i}} \sigma_i^\ell\right|\right] \geq \sqrt{\frac{2^{4^i}}{2}}.$$

Thus, $\mathbb{E}[R_i] \geq \sqrt{\frac{2^{4^i}}{8}}$. Then consider the following sequence: $Z_{i,j} = \mathbb{E}[R_i | Y_{\leq T_{i-1}+j}]$. Notice that for every $i$, for every $j \leq T_i - T_{i-1}$, we have:

$$\mathbb{E}[|Z_{i,j}|] \leq \mathbb{E}[|\mathbb{E}[R_i | Y_{\leq T_{i-1}+j}]|] \leq \mathbb{E}[\mathbb{E}[|R_i| \,|\, Y_{\leq T_{i-1}+j}]] = \mathbb{E}[|R_i|] \leq T_i - T_{i-1} = 2^{4^i} < \infty.$$

and

$$\mathbb{E}[Z_{i,j+1}|Y_{T_{i-1}+1}, \ldots, Y_{T_{i-1}+j}] = \mathbb{E}[\mathbb{E}[R_i|Y_{\leq T_{i-1}+j+1}]|Y_{T_{i-1}+1}, \ldots, Y_{T_{i-1}+j}]$$
$$= \mathbb{E}[R_i|Y_{T_{i-1}+1}, \ldots, Y_{T_{i-1}+j}] = \mathbb{E}[R_i|Y_{\leq T_{i-1}+j}] = Z_{i,j}.$$

Therefore, for every $i$, the sequence $Z_{i,j}$ is a martingale indexed by $j$. By the definition of regret, we have $-1 \leq Z_{i,j+1} - Z_{i,j} \leq 1$. Then we have

$$\mathbb{P}\left[R_i \leq \sqrt{\frac{2^{4^i}}{32}}\right] = \mathbb{P}\left[R_i \leq \sqrt{\frac{2^{4^i}}{8}} - \sqrt{\frac{2^{4^i}}{32}}\right]$$
$$\leq \mathbb{P}\left[R_i \leq \mathbb{E}[R_i] - \sqrt{\frac{2^{4^i}}{32}}\right]$$
$$\leq e^{-\frac{1}{16}}$$

The last inequality comes from Azuma's Inequality. Thus, we have

$$\mathbb{P}\left[R_i \geq \sqrt{\frac{2^{4^i}}{32}}\right] \geq 1 - e^{-\frac{1}{16}},$$

for every $i$. Thus, for every $i$, if we choose $h_i^* = \arg\min_{h \in \{h_1, h_2\}} \left(\sum_{t=T_{i-1}+1}^{T_i} \mathbb{I}[Y_t \neq h(X_t)]\right)$ to generate $Y^{*,i}$, i.e., for every $t$, $Y_t^{*,i} = h_i^*(x)$, we will have

$$\text{Regret}(\mathcal{A}, (\mathbb{X}, \mathbb{Y}, \mathbb{Y}^{*,i}), T_i) \geq \sqrt{\frac{2^{4^i}}{32}} - T_{i-1} \geq 2^{4^{i-1}}\left(2^{4^{i-1} - \frac{5}{2}} - 2\right) = \Omega(\sqrt{T_i}),$$

with probability at least $1 - e^{-\frac{1}{16}}$. Then because $h_i^* = h_1$ or $h_2$, due to pigeonhole principle, there is an $h^*$, such that there are infinitely many $i$, $h_i^* = h^*$. Let $Y_t^* = h^*(x)$ for every $t$. Let event $E_i$ be that $\text{Regret}(\mathcal{A}, (\mathbb{X}, \mathbb{Y}, \mathbb{Y}^*), T_i) = \Omega(\sqrt{T_i})$. By the definition of the limit of superior of events, the fact that the indicator function is either 0 or 1 and the reversed Fatou's lemma, we have

$$\mathbb{P}\left[\limsup_{i \to \infty} E_i\right] = \mathbb{E}\left[\limsup_{i \to \infty} \mathbb{I}[E_i \text{ happens.}]\right] \geq \limsup_{i \to \infty} \mathbb{E}[\mathbb{I}[E_i \text{ happens.}]] \geq 1 - e^{-\frac{1}{16}}.$$

Thus,

$$\limsup_{i \to \infty} \frac{1}{\sqrt{T_i}} \text{Regret}(\mathcal{A}, (\mathbb{X}, \mathbb{Y}, \mathbb{Y}^*), T_i) = c$$

for some constant $c > 0$ with probability greater than 0. Therefore,

$$\limsup_{T \to \infty} \frac{1}{\sqrt{T}} \text{Regret}(\mathcal{A}, (\mathbb{X}, \mathbb{Y}, \mathbb{Y}^*), T) \geq \limsup_{i \to \infty} \frac{1}{\sqrt{T_i}} \text{Regret}(\mathcal{A}, (\mathbb{X}, \mathbb{Y}, \mathbb{Y}^*), T_i) = c$$

with probability greater than 0. Thus, for any learning algorithm $\mathcal{A}$, there is a deterministic sequence $(\mathbb{X}, \mathbb{Y}, \mathbb{Y}^*)$ such that $\text{Regret}(\mathcal{A}, (\mathbb{X}, \mathbb{Y}, \mathbb{Y}^*), T) \neq o(\sqrt{T})$. That finishes the proof. □

## D. Learnability when the Stochastic Process is Known

In this section, we discuss the universal online learnability when the stochastic process generating the instance sequence is known to the learner instead of the instance sequence. This setting is equivalent to the condition when all processes admit universal online learning as asked in the work of Hanneke & Wang (2024). As we introduced stochastic processes in this model, several definitions are slightly different from the original transductive online learning with deterministic sequences. We define the changed definition for the stochastic process case again here.

Let $(\mathcal{X}, \mathcal{B})$ be a measurable space, where $\mathcal{X}$ is a non-empty set and $\mathcal{B}$ is a Borel $\sigma$-algebra generated by the separable metrizable topology $\mathcal{T}$. This is called *instance space*. And $\mathcal{Y}$ is also a non-empty measurable space called *label space*. Here

we also focus on the learning on 0-1 loss. A stochastic process $\mathbb{X} = \{X_t\}_{t \in \mathbb{N}}$ is a sequence of $\mathcal{X}$-valued random variables, which is called instance process and a stochastic process $\mathbb{Y} = \{Y_t\}_{t \in \mathbb{N}}$ is a sequence of $\mathcal{Y}$-valued random variables, which is called label process. The concept class $\mathcal{H} \subseteq \mathcal{Y}^{\mathcal{X}}$ is a non-empty set of measurable functions from $\mathcal{X}$ to $\mathcal{Y}$. In order to avoid the complicated discussion of the measurability issue, we assume the instance space $\mathcal{X}$ and the label space $\mathcal{Y}$ are both countable (countably infinite).

We also need to redefine the online learning algorithm for the stochastic process setting, which is as follows. The online learning rule is a sequence of measurable functions: $f_t : \mathcal{X}^{t-1} \times \mathcal{Y}^{t-1} \times \mathcal{X} \to \mathcal{Y}$, where $t$ is a non-negative integer. For convenience, we also define $\hat{h}_{t-1} = f_t(X_{<t}, Y_{<t})$, here $(X_{<t}, Y_{<t}) = \{(X_i, Y_i)\}_{i<t}$ is the history before round $t$.

Then we need to define the realizable setting, which follows the tradition of universal learning.

**Definition D.1.** For every concept class $\mathcal{H}$, we can define the following set of processes $\mathrm{R}(\mathcal{H})$:

$$\mathrm{R}(\mathcal{H}) := \left\{ (\mathbb{X}, \mathbb{Y}) = \{(X_i, Y_i)\}_{i \in \mathbb{N}} : \text{with probability } 1, \forall n < \infty, \{(X_i, Y_i)\}_{i \leq n} \text{ realizable by } \mathcal{H} \right\}.$$

In the same way, the set of realizable label processes:

**Definition D.2.** For every concept class $\mathcal{H}$ and data process $\mathbb{X}$, define a set $\mathrm{R}(\mathcal{H}, \mathbb{X})$ of label processes:

$$\mathrm{R}(\mathcal{H}, \mathbb{X}) := \left\{ \mathbb{Y} = \{Y_i\}_{i \in \mathbb{N}} : (\mathbb{X}, \mathbb{Y}) \in \mathrm{R}(\mathcal{H}) \text{ and } \exists \text{ a non-random function } f \text{ s.t. } Y_i = f(X_i) \right\}.$$

In other words, $\mathrm{R}(\mathcal{H}, \mathbb{X})$ are label processes $\mathbb{Y} = f(\mathbb{X})$ s.t. $(\mathbb{X}, f(\mathbb{X})) \in \mathrm{R}(\mathcal{H})$. Importantly, while every $f \in \mathcal{H}$ satisfies $f(\mathbb{X}) \in \mathrm{R}(\mathcal{H}, \mathbb{X})$, there can exist $f \notin \mathcal{H}$ for which this is also true, due to $\mathrm{R}(\mathcal{H})$ only requiring realizable *prefixes* (thus, in a sense, $\mathrm{R}(\mathcal{H}, \mathbb{X})$ represents label sequences by functions in a *closure* of $\mathcal{H}$ defined by $\mathbb{X}$).[7]

Then we define the universal consistency under $\mathbb{X}$ and $\mathcal{H}$ in the realizable case. An online learning rule is universally consistent under $\mathbb{X}$ and $\mathcal{H}$ if its long-run average loss approaches 0 almost surely for all realizable label processes. Formally,

**Definition D.3.** An online learning rule is *universally consistent* under $\mathbb{X}$ and $\mathcal{H}$ for the realizable case, if for *every* $\mathbb{Y} \in \mathrm{R}(\mathcal{H}, \mathbb{X})$, $\limsup_{T \to \infty} \frac{1}{T} \sum_{t=1}^{T} \mathbb{I}\left[Y_t \neq \hat{h}_{t-1}(X_t)\right] = 0$ a.s.

We also define the universal consistency under $\mathbb{X}$ and $\mathcal{H}$ for the agnostic case. Here, we release the restrictions that $\mathbb{Y} \in \mathrm{R}(\mathcal{H}, \mathbb{X})$; instead, the label process $\mathbb{Y}$ can be set in any possible way, even dependent on the history of the algorithm's predictions. Thus, the average loss may be linear and inappropriate for defining consistency. Therefore, we compare the performance of our algorithm with the performance of the best possible $\mathbb{Y}^* \in \mathrm{R}(\mathcal{H}, \mathbb{X})$, which is usually referred to as *regret*. We say an online algorithm is universally consistent under $\mathbb{X}$ and $\mathcal{H}$ for the agnostic case if its long-run average regret is low for every label process. Formally,

**Definition D.4.** An online learning rule is *universally consistent* under $\mathbb{X}$ and $\mathcal{H}$ for the agnostic case, if for *every* $\mathbb{Y}^* \in \mathrm{R}(\mathcal{H}, \mathbb{X})$ and for *every* $\mathbb{Y}$, $\limsup_{n \to \infty} \frac{1}{n} \sum_{t=1}^{n} \left( \mathbb{I}\left[Y_t \neq \hat{h}_{t-1}(X_t)\right] - \mathbb{I}\left[Y_t \neq Y_t^*\right] \right) \leq 0$ a.s.

Then we provide the following definition to describe our main theorem in this section.

**Definition D.5.** We say a process $\mathbb{X}$ admits *universal online learning* if there exists an online learning rule that is universally consistent under $\mathbb{X}$ and $\mathcal{H}$.

Then comes the main theorem of this section.

**Theorem D.6.** All *processes admit universal online learning, if and only if $\mathcal{H}$ has no infinite indifferent LCLL tree.*

The necessity of this theorem is shown as Lemma 4.1. Then we only need to prove the sufficiency. The proof of the sufficiency is very similar to the proof of Lemma 4.2. The algorithm also contains two stages: first, we update the LCLL game, and after a finite number of updates, we reach a condition where with probability 1, the length of the longest subsequences that are shattered by the concept class $\mathcal{H}'$ is at most $d$. Second, we design a learning algorithm that can learn the concept class $\mathcal{H}'$ with at most $o(T)$ number of mistakes. Algorithm 5 is our algorithm. The only change from Algorithm 1 is the prediction rule. We modify the prediction rule such that it suits the stochastic process setting.

As the first stage of the algorithm is exactly the same, Lemma 4.5 still holds. We only need to show that we can learn the partial concept with $o(T)$ number of mistakes, which is Lemma D.7.

---

[7]For instance, for $\mathcal{X} = \mathbb{N}$, for the process $X_i = i$, and for $\mathcal{H} = \{\mathbb{1}_{\{i\}} : i \in \mathcal{X}\}$ (singletons), the all-0 sequence is in $\mathrm{R}(\mathcal{H}, \mathbb{X})$ though the all-0 function is not in $\mathcal{H}$.

---

**Algorithm 5** Learning algorithm from winning strategy

---

$k = 1, U = \{\}, t' \leftarrow 0, t_0 \leftarrow 0$.

**for** $t = 1, 2, 3, \ldots$ **do**

    **if** $\exists t' \leq t_1 < t_2 < \cdots < t_k \leq t$ and $C_k = \{(y_{t_{k-1}^{k-1}+1,\varnothing}, \ldots, y_{t_k^1,\varnothing}^{u_1}, y_{t_k^1+1,u_1}, \ldots, y_{t_k^2,u_1}^{u_2} \ldots, y_{t_k^k,u_{<k}}^{u_k}) : u \in \{0,1\}^k\}$

    such that $g_U((t_1, \ldots, t_k), C_k) = (Y_{t'}, \ldots, Y_t)$ **then**

        Advance the game:

        $U \leftarrow U \cup \{((t_1, \ldots, t_k, C_k), g_U((t_1, \ldots, t_k), C_k))\}$.

        $k \leftarrow k + 1$.

        $L \leftarrow \emptyset$.

        $m \leftarrow 1$.

        $t' \leftarrow t$.

    **end if**

    Predict

$$\hat{Y}_t = \arg\min_y \Pr\left[w(\mathcal{H}_{L \cup (X_t, y)}^{g_U}, X_{[t, t(m)+t']}) \leq \frac{1}{2} w(\mathcal{H}_L^{g_U}, X_{[t, t(m)+t']}) \,\middle|\, X_{\leq t}\right]$$

    **if** $Y_t \neq \hat{Y}_t$ **then**

        $L \leftarrow L \cup \{(X_t, Y_t)\}$.

    **end if**

    **if** $t \geq \frac{m(m+1)}{2} + t'$. **then**

        $m \leftarrow m + 1$.

    **end if**

**end for**

---

**Algorithm 6** Subroutine for learning under the condition that the length of the longest subsequence shattered by $\mathcal{H}$ is at most $d$.

---

$L \leftarrow \emptyset$.

$m \leftarrow 1$.

**for** $t = 1, 2, 3, \ldots$ **do**

    Predict

$$\hat{Y}_t = \arg\min_y \Pr\left[w(\mathcal{H}_{L \cup (X_t, y)}, X_{[t, t(m)]}) \leq \frac{1}{2} w(\mathcal{H}_L, X_{[t, t(m)]}) \,\middle|\, X_{\leq t}\right]$$

    **if** $Y_t \neq \hat{Y}_t$ **then**

        $L \leftarrow L \cup \{(X_t, Y_t)\}$.

    **end if**

    **if** $t \geq \frac{m(m+1)}{2}$. **then**

        $m \leftarrow m + 1$.

    **end if**

**end for**

---

**Lemma D.7.** *For any process* $\mathbb{X}$*, if the length of the longest subsequence shattered by* $\mathcal{H}'$ *is at most* $d$ *with probability* $1$*, Algorithm 6 only makes* $o(T)$ *mistakes almost surely when* $T \to \infty$*.*

To prove the property of the algorithm, we first need a lemma about the prediction rule.

**Lemma D.8.** *If* $w(\mathcal{H}', X_{\leq T}) = |\{S : S \subseteq \{X_i\}_{i \leq T} \text{ such that } S \text{ shattered by } \mathcal{H}'\}|$*, for every sequence* $\mathbb{X}$*, and every partial concept* $\mathcal{H}$*, we have at most one* $y$ *such that*

$$\Pr\left[w(\mathcal{H}_{L \cup (X_t, y)}, X_{[t, t(m)]}) \leq \frac{1}{2} w(\mathcal{H}_L, X_{[t, t(m)]}) \,\Big|\, X_{\leq t}\right] < \frac{1}{2}.$$

*Proof.* We can prove this lemma by contradiction. First, we have the following fact: For every pair $(y, y^*)$ such that $y \neq y^*$, and every sequence $\mathbb{X}$, we have:

$$w(\mathcal{H}_{L \cup (X_t, y^*)}, X_{[t, t(m)]}) + w(\mathcal{H}_{L \cup (X_t, y)}, X_{[t, t(m)]}) \leq w(\mathcal{H}_L, X_{[t, t(m)]}).$$

This is because if a $k'$ tuple $(X_{t_1}, \ldots, X_{t_{k'}})$ is shattered by $\mathcal{H}_{L \cup (X_t, y^*)}$ and $\mathcal{H}_{L \cup (X_t, y)}$, the $k' + 1$ tuple $(X_t, X_{t_1}, \ldots, X_{t_{k'}})$ will be shattered by $\mathcal{H}_L$. Thus, every tuple that contributes twice in the left part will also contribute twice in the right part. Therefore, the inequality holds.

Then, suppose there are $y', y'', y' \neq y''$, such that,

$$\Pr\left[w(\mathcal{H}_{L \cup (X_t, y')}, X_{[t, t(m)]}) \leq \frac{1}{2} w(\mathcal{H}_L, X_{[t, t(m)]}) \,\Big|\, X_{\leq t}\right] < \frac{1}{2}.$$

and

$$\Pr\left[w(\mathcal{H}_{L \cup (X_t, y'')}, X_{[t, t(m)]}) \leq \frac{1}{2} w(\mathcal{H}_L, X_{[t, t(m)]}) \,\Big|\, X_{\leq t}\right] < \frac{1}{2}.$$

Thus, with probability greater than 0, we have

$$w(\mathcal{H}_{L \cup (X_t, y')}, X_{[t, t(m)]}) > \frac{1}{2} w(\mathcal{H}_L, X_{[t, t(m)]}),$$

and

$$w(\mathcal{H}_{L \cup (X_t, y'')}, X_{[t, t(m)]}) > \frac{1}{2} w(\mathcal{H}_L, X_{[t, t(m)]}).$$

Thus,

$$w(\mathcal{H}_{L \cup (X_t, y')}, X_{[t, t(m)]}) + w(\mathcal{H}_{L \cup (X_t, y'')}, X_{[t, t(m)]}) > w(\mathcal{H}_L, X_{[t, t(m)]}).$$

That is a contradiction, and we finish the proof. $\square$

Then we can start to prove Lemma D.7, though the proof is very similar to the proof in the work of Hanneke & Wang (2024), we provide it for completeness.

*Proof of Lemma D.7.* As we defined, the weight function, $w(\mathcal{H}', X_{\leq T}) = |\{S : S \subseteq \{X_i\}_{i \leq T} \text{ such that } S \text{ shattered by } \mathcal{H}'\}|$, which is the number of the subsequences of the sequence $X_{\leq T}$ that can be shattered by the concept class $\mathcal{H}'$.

Consider the $k$-th batch, consisting of $W_k = \{X_{\frac{k(k-1)}{2}+1}, \cdots, X_{\frac{k(k+1)}{2}}\}$. Let

$$Z_k = \sum_{t=\frac{k(k-1)}{2}+1}^{\frac{k(k+1)}{2}} \mathbb{I}\left[\hat{Y}_t \neq Y_t\right],$$

and

$$V_k = Z_k - \mathbb{E}\left[Z_k \,\Big|\, X_{\leq \frac{k(k-1)}{2}}\right].$$

Notice that

$$\mathbb{E}\left[V_k \,\Big|\, X_{\leq \frac{k(k-1)}{2}}\right]$$
$$= \mathbb{E}\left[Z_k - \mathbb{E}\left[Z_k \,\Big|\, X_{\leq \frac{k(k-1)}{2}}\right] \,\Big|\, X_{\leq \frac{k(k-1)}{2}}\right]$$
$$= \mathbb{E}\left[Z_k \,\Big|\, X_{\leq \frac{k(k-1)}{2}}\right] - \mathbb{E}\left[Z_k \,\Big|\, X_{\leq \frac{k(k-1)}{2}}\right] = 0. (a.s.)$$

Thus, the sequence $V_k$ is a martingale difference sequence with respect to the block sequence, $W_1, W_2, \cdots$. By the definition of $V_k$, we also have $-k \leq V_k \leq k$. Then by Azuma's Inequality, with probability $1 - \frac{1}{K^2}$, we have

$$\sum_{k=1}^K Z_k \leq \sum_{k=1}^K \mathbb{E}\left[Z_k \,\Big|\, X_{\leq \frac{k(k-1)}{2}}\right] + \sqrt{-\log\left(\frac{1}{K^2}\right) \cdot 2 \cdot \left(\sum_{k=1}^K k^2\right)}$$
$$\leq \sum_{k=1}^K \mathbb{E}\left[Z_k \,\Big|\, X_{\leq \frac{k(k-1)}{2}}\right] + \sqrt{4K^3 \log K}.$$

Then we need to get an upper bound for $\mathbb{E}\left[Z_k \,\Big|\, X_{\leq \frac{k(k-1)}{2}}\right]$. According to the prediction rule and Lemma D.8, every time we make a mistake, we have

$$\Pr\left[w(\mathcal{H}_{L \cup (X_t, Y_t)}, X_{[t, \frac{k(k+1)}{2}]}) \leq \frac{1}{2} w(\mathcal{H}_L, X_{[t, \frac{k(k+1)}{2}]}) \,\Big|\, X_{\leq t}\right] \geq \frac{1}{2}. \tag{4}$$

Due to the linearity of expectation, for every $k$,

$$\mathbb{E}\left[\sum_{t=\frac{k(k-1)}{2}}^{\frac{k(k+1)}{2}} \mathbb{I}\left[\hat{Y}_t \neq Y_t\right] \,\Bigg|\, X_{\leq \frac{k(k-1)}{2}}\right]$$

$$= \mathbb{E}\left[\sum_{t=\frac{k(k-1)}{2}}^{\frac{k(k+1)}{2}} \mathbb{I}\left[\hat{Y}_t \neq Y_t\right] \mathbb{I}\left[w(\mathcal{H}_{L_t}, X_{[t+1, \frac{k(k+1)}{2}]}) \leq \frac{1}{2} w(\mathcal{H}_{L_{t-1}}, X_{[t, \frac{k(k+1)}{2}]})\right] \,\Bigg|\, X_{\leq \frac{k(k-1)}{2}}\right]$$

$$+ \mathbb{E}\left[\sum_{t=\frac{k(k-1)}{2}}^{\frac{k(k+1)}{2}} \mathbb{I}\left[\hat{Y}_t \neq Y_t\right] \mathbb{I}\left[w(\mathcal{H}_{L_t}, X_{[t+1, \frac{k(k+1)}{2}]}) > \frac{1}{2} w(\mathcal{H}_{L_{t-1}}, X_{[t, \frac{k(k+1)}{2}]})\right] \,\Bigg|\, X_{\leq \frac{k(k-1)}{2}}\right].$$

Here $L_t = \{(X_i, Y_i)\}$, where $i \leq t$ and the algorithm makes a mistake at round $i$.

Notice the first part is the expected number of mistakes, each of which decreases the weight by half. For every realization of $X_{[\frac{k(k-1)}{2}, \frac{k(k+1)}{2}]}, x_{[\frac{k(k-1)}{2}, \frac{k(k+1)}{2}]}$, let

$$u(k) = \sum_{i=\frac{k(k-1)}{2}}^{\frac{k(k+1)}{2}} \mathbb{I}\left[\hat{Y}_t \neq Y_t\right] \mathbb{I}\left[w(\mathcal{H}_{L_t}, x_{[t+1, \frac{k(k+1)}{2}]}) \leq \frac{1}{2} w(\mathcal{H}_{L_{t-1}}, x_{[t, \frac{k(k+1)}{2}]})\right].$$

By the definition of the weight function and the fact that the length of the longest subsequence shattered by $\mathcal{H}'$ is at most $d$, $w(\mathcal{H}_{L_{\frac{k(k-1)}{2}}}, x_{[\frac{k(k-1)}{2}, \frac{k(k+1)}{2}]}) \leq k^d$. Consider the last round $t \leq \frac{k(k+1)}{2}$ that $\hat{Y}_t \neq Y_t$, we have $w(\mathcal{H}_{L_{t-1}, x_{[t, \frac{k(k+1)}{2}]}}) \geq 1$, as $\{x_t\}$ must be shattered. Thus, we have $2^{u(k)-1} w(\mathcal{H}_{L_{t-1}}, x_{[t, \frac{k(k+1)}{2}]}) \leq w(\mathcal{H}, x_{[\frac{k(k-1)}{2}, \frac{k(k+1)}{2}]})$. Therefore, $u(k) \leq d \log k + 1$, for every realization, $x_{[\frac{k(k-1)}{2}, \frac{k(k+1)}{2}]}$. Thus,

$$\mathbb{E}\left[\sum_{t=\frac{k(k-1)}{2}}^{\frac{k(k+1)}{2}} \mathbb{I}\left[\hat{Y}_t \neq Y_t\right] \mathbb{I}\left[w(\mathcal{H}_{L_t}, X_{[t+1, \frac{k(k+1)}{2}]}) \leq \frac{1}{2} w(\mathcal{H}_{L_{t-1}}, X_{[t, \frac{k(k+1)}{2}]})\right] \,\Bigg|\, X_{\leq \frac{k(k-1)}{2}}\right] \leq 2d \log k. \tag{5}$$

Then consider the second part, we have

$$\mathbb{E}\left[\sum_{t=\frac{k(k-1)}{2}}^{\frac{k(k+1)}{2}} \mathbb{I}\left[\hat{Y}_t \neq Y_t\right] \mathbb{I}\left[w(\mathcal{H}_{L_t}, X_{[t+1, \frac{k(k+1)}{2}]}) > \frac{1}{2}w(\mathcal{H}_{L_{t-1}}, X_{[t, \frac{k(k+1)}{2}]})\right] \middle| X_{\leq \frac{k(k-1)}{2}}\right]$$

$$= \mathbb{E}\left[\mathbb{E}\left[\sum_{t=\frac{k(k-1)}{2}}^{\frac{k(k+1)}{2}} \mathbb{I}\left[\hat{Y}_t \neq Y_t\right] \mathbb{I}\left[w(\mathcal{H}_{L_t}, X_{[t+1, \frac{k(k+1)}{2}]}) > \frac{1}{2}w(\mathcal{H}_{L_{t-1}}, X_{[t, \frac{k(k+1)}{2}]})\right] \middle| X_{\leq t}\right] \middle| X_{\leq \frac{k(k-1)}{2}}\right]$$

$$= \mathbb{E}\left[\sum_{t=\frac{k(k-1)}{2}}^{\frac{k(k+1)}{2}} \mathbb{I}\left[\hat{Y}_t \neq Y_t\right] \mathbb{E}\left[\mathbb{I}\left[w(\mathcal{H}_{L_t}, X_{[t+1, \frac{k(k+1)}{2}]}) > \frac{1}{2}w(\mathcal{H}_{L_{t-1}}, X_{[t, \frac{k(k+1)}{2}]})\right] \middle| X_{\leq t}\right] \middle| X_{\leq \frac{k(k-1)}{2}}\right]$$

This is because $\hat{Y}_t$ and $Y_t$ only depend on $X_{\leq t}$. Due to the equation 4, we have

$$\mathbb{I}\left[\hat{Y}_t \neq Y_t\right] \mathbb{E}\left[\mathbb{I}\left[w(\mathcal{H}_{L_t}, X_{[t+1, \frac{k(k+1)}{2}]}) > \frac{1}{2}w(\mathcal{H}_{L_{t-1}}, X_{[t, \frac{k(k+1)}{2}]})\right] \middle| X_{\leq t}\right]$$

$$= \mathbb{I}\left[\hat{Y}_t \neq Y_t\right] \Pr\left[w(\mathcal{H}_{L_t}, X_{[t+1, \frac{k(k+1)}{2}]}) > \frac{1}{2}w(\mathcal{H}_{L_{t-1}}, X_{[t, \frac{k(k+1)}{2}]}) \middle| X_{\leq t}\right]$$

$$\leq \frac{1}{2}\mathbb{I}\left[\hat{Y}_t \neq Y_t\right].$$

Thus,

$$\mathbb{E}\left[\sum_{t=\frac{k(k-1)}{2}}^{\frac{k(k+1)}{2}} \mathbb{I}\left[\hat{Y}_t \neq Y_t\right] \mathbb{I}\left[w(\mathcal{H}_{L_t}, X_{[t+1, \frac{k(k+1)}{2}]}) > \frac{1}{2}w(\mathcal{H}_{L_{t-1}}, X_{[t, \frac{k(k+1)}{2}]})\right] \middle| X_{\leq \frac{k(k-1)}{2}}\right] \tag{6}$$

$$\leq \frac{1}{2}\mathbb{E}\left[\sum_{t=\frac{k(k-1)}{2}}^{\frac{k(k+1)}{2}} \mathbb{I}\left[\hat{Y}_t \neq Y_t\right] \middle| X_{\leq \frac{k(k-1)}{2}}\right]$$

Combining these two inequalities (5 and 6), we have

$$\mathbb{E}\left[\sum_{t=\frac{k(k-1)}{2}}^{\frac{k(k+1)}{2}} \mathbb{I}\left[\hat{Y}_t \neq Y_t\right] \middle| X_{\leq \frac{k(k-1)}{2}}\right] \leq 2d\log k + \frac{1}{2}\mathbb{E}\left[\sum_{t=\frac{k(k-1)}{2}}^{\frac{k(k+1)}{2}} \mathbb{I}\left[\hat{Y}_t \neq Y_t\right] \middle| X_{\leq \frac{k(k-1)}{2}}\right]. \tag{7}$$

Thus, for any $k$, we have

$$\mathbb{E}\left[\sum_{t=\frac{k(k-1)}{2}}^{\frac{k(k+1)}{2}} \mathbb{I}\left[\hat{Y}_t \neq Y_t\right] \middle| X_{\leq \frac{k(k-1)}{2}}\right] \leq 4d\log k. \tag{8}$$

According to the inequality 8 for every $k$, $\mathbb{E}\left[Z_k \middle| X_{\leq \frac{k(k-1)}{2}}\right] \leq 4d\log k$. Thus, with probability at least $1 - \frac{1}{K^2}$,

$$\sum_{k=1}^{K} Z_k \leq \sum_{k=1}^{K} 4d\log k + \sqrt{4K^3 \log K} \leq 4dK\log K + \sqrt{4K^3 \log K}.$$

By the definition of $Z_k$, with probability at least $1 - \frac{1}{K^2}$,

$$\sum_{t=1}^{\frac{K(K+1)}{2}} \mathbb{I}\left[\hat{Y}_t \neq Y_t\right] \leq 4dK\log K + \sqrt{4K^3 \log K} \leq (4d+2)\sqrt{K^3 \log K}. \tag{9}$$

Let $T_K = \frac{K(K+1)}{2}$ be the number of instances in the sequence, with probability at least $1 - \frac{1}{K^2}$

$$\sum_{t=1}^{T_K} \mathbb{I}\left[\hat{Y}_t \neq Y_t\right] \leq (4d+2)T_K^{\frac{3}{4}}\sqrt{\frac{1}{2}\log T_K}. \tag{10}$$

Define the event $E_K$ as the event that in the sequence $X_{\leq T_K}$, $\sum_{t=1}^{T_K} \mathbb{I}\left[\hat{Y}_t \neq Y_t\right] > (4d+2)T_K^{\frac{3}{4}}\sqrt{\frac{1}{2}\log T_K}$. Then we know $\Pr[E_K] \leq \frac{1}{K^2}$. Notice the fact that for any $K \in \mathbb{N}$, $\sum_{k=1}^{K} \frac{1}{k^2} \leq \frac{\pi^2}{6}$. By Borel-Cantelli lemma, we know that for any $T_K = \frac{K(K+1)}{2}$ large enough, $\sum_{t=1}^{T_K} \mathbb{I}\left[\hat{Y}_t \neq Y_t\right] \leq (4d+2)T_K^{\frac{3}{4}}\sqrt{\frac{1}{2}\log T_K}$ happens with probability 1.

Then for any large enough $T$, we have $T_K \leq T \leq T_{K+1} \leq 2T$. Thus, with probability 1,

$$\sum_{t=1}^{T} \mathbb{I}\left[\hat{Y}_t \neq Y_t\right] \leq (4d+2)T_{K+1}^{\frac{3}{4}}\sqrt{\frac{1}{2}\log T_{K+1}} \tag{11}$$

$$\leq (4d+2)(2T)^{\frac{3}{4}}\sqrt{\frac{1}{2}\log 2T}. \tag{12}$$

Therefore, for any large enough $T$ and a universal constant $c$, with probability 1,

$$\sum_{t=1}^{T} \mathbb{I}\left[\hat{Y}_t \neq Y_t\right] \leq cT^{\frac{3}{4}}\sqrt{\log T}. \tag{13}$$

Notice that $cT^{\frac{3}{4}}\sqrt{\log T}$ is $o(T)$. That finishes the proof. $\square$

We can also extend the learnability results to the agnostic case. Formally,

**Theorem D.9.** *If $\mathcal{H}$ has no infinite indifferent LCLL tree, all processes admit universal online learning in the agnostic setting.*

The proof of this theorem is very similar to the proof in Section C. The only difference is that the experts simulate the running of Algorithm 5 instead of Algorithm 1. For completeness, we provide the expert as Algorithm 7.

Thus, we need to show we can still build the sequence $j_T = o(T)$ such that for large enough $T$ there exists $j < j_T$, for every $t < T$, $e_{i,j}(X_t) = Y_t^*$ for at most $o(T)$ times. Formally,

**Lemma D.10.** *If $\mathcal{H}$ has no infinite indifferent LCLL tree, for every realizable sequence $(\mathbb{X}, \mathbb{Y}) \in R(\mathcal{H})$, we have a sequence $\{j_T\}_{T \in \mathbb{N}}$ satisfies $\log j_T = o(T)$, such that for every large enough time $T$, we have an expert $e_{i,j}$ with $j \leq j_T$, such that for all $t \leq T$, $Y_t = e_{i,j}(X_t)$ except for at most $o(T)$ times.*

The proof of this lemma is also similar to the proof of Lemma C.1. The only difference is that we only know that for every $T$, the number of mistakes made by Algorithm 5 is $o(T)$ instead of $O(\log T)$. Therefore, we need to use the trick from the work of Hanneke & Wang (2024), we get the index $j$ of Expert $I, J$, through the following order. We order it by the value of $|J| \max J$ (use $|J|$ as tie breaking). For example, $J = \{2, 3, 5\}$, the value used to index it is $|J| \max J = 3 \cdot 5 = 15$. Therefore, we have the upper bound of $j_T$ as follows.

$$j_T \leq |\{J : |J| \max J \leq k\}| = 1 + \sum_{m=1}^{k} |\{J : |J| \leq \frac{k}{m}, \max J = m\}|$$

$$= 1 + \sum_{m=1}^{\sqrt{k}} 2^{m-1} + \sum_{m=\sqrt{k}}^{k} \binom{m-1}{\leq (\frac{k}{m} - 1)} \leq 2^{\sqrt{k}} + \sum_{m=\sqrt{k}}^{k} \left(\frac{em^2}{k}\right)^{\frac{k}{m}} \leq (k+1)e^{\sqrt{k}}.$$

Thus, we have $\log j_T \leq \sqrt{|J|T}$. We know $|J| = o(T)$, thus, $\log j_T = o(T)$. And there is $j < j_T$ such that for all $t < T$, $e_{i,j}(X_t) \neq Y_t$ if and only if $j \in J$. That finishes the proof.

---

**Algorithm 7** Expert $I, J$ (indexed by $e_{i,j}$).

---

$U \leftarrow \{\}. \ t' \leftarrow 0. k \leftarrow 1. L \leftarrow \{\}. \ m \leftarrow 1$

**for** $t = 1, 2, 3, \ldots$ **do**

  **if** $t \leq k$ **then**

    **if** $\exists t' \leq t_1 < t_2 < \cdots < t_k \leq t$ and $C_k = \{(y_{t_{k-1}^{k-1}+1,\varnothing}, \ldots, y_{t_k^1,\varnothing}^{u_1}, y_{t_k^1+1,u_1}, \ldots, y_{t_2^2,u_1}^{u_2} \ldots, y_{t_k^k,u_{<k}}^{u_k}) : u \in$

    $\{0,1\}^k\}$ such that $g_U((t_1, \ldots, t_k), C_k) = (Y_{t'}, \ldots, Y_t)$ **then**

      Advance the game:

      $U \leftarrow U \cup \{((t_1, \ldots, t_k, C_k), g_U((t_1, \ldots, t_k), C_k))\}$.

      $k \leftarrow k + 1$.

      $t' \leftarrow t$.

    **end if**

    Predict $\hat{Y}_t = Y_t^*$.

  **else**

    Predict

$$\hat{Y}_t = \arg\min_y \Pr\left[w(\mathcal{H}_{L \cup (X_t, y)}^{g_U}, X_{[t, t(m)+t']}) \leq \frac{1}{2} w(\mathcal{H}_L^{g_U}, X_{[t, t(m)+t']}) \,\middle|\, X_{\leq t}\right]$$

    .

    **if** $t \in J$ **then**

      $L \leftarrow L \cup \{(X_t, Y_t)\}$.

    **end if**

    **if** $t \geq \frac{m(m+1)}{2} + t'$. **then**

      $m \leftarrow m + 1$.

    **end if**

  **end if**

**end for**

---

Thus, the regret of the algorithm above is

$$\text{Regret}(\mathcal{A}, (\mathbb{X}, \mathbb{Y}, \mathbb{Y}^*), T) = O\left(\sqrt{T \log\log T + T(\log i + \log j) + (\log i + \log j)}\right) + o(T)$$

$$= O\left(\sqrt{T \log\log T + T \log j_T + \log j_T)}\right) + o(T)$$

$$= O\left(\sqrt{T \log\log T + T o(T) + o(T))}\right) + o(T) = o(T).$$

Lemma 4.1 also shows the necessity of Theorem D.9.

## D.1. Discussion

This part provides more intuition about the optimistically universal online learning problem. The term "optimistically universal online learning" was introduced by Hanneke (2021) in order to understand the learnability under the *minimal assumptions*, that is, only assume that the data processes admit universal online learning. Intuitively, we would like to ask whether it is possible to learn whenever learning is possible. More recently, the work of Hanneke & Wang (2024) adds the concept class restriction into the optimistically universal online learning setting and fully characterizes this problem for binary classification. In their work, they first figure out the minimal assumption that a process admits the universal online learning. Then they figure out whether there is a learning algorithm that learns all processes that admit universal online learning. We have defined admitting universal online learning before, so we define optimistically universally online learnable here. Formally,

**Definition D.11.** An online learning rule is *optimistically universal* under concept class $\mathcal{H}$ if it is strongly universally consistent under every process $\mathbb{X}$ that admits strongly universally consistent online learning under concept class $\mathcal{H}$.

If there is an online learning rule that is *optimistically universal* under concept class $\mathcal{H}$, we say $\mathcal{H}$ is *optimistically universally online learnable*.

According to the work of Hanneke et al. (2023c), we have the following theorem about the optimistically universal online learnability when all processes admit universal online learning.

**Theorem D.12** ((Hanneke et al., 2023c)). *If and only if $\mathcal{H}$ has no infinite Littlestone tree, $\mathcal{H}$ is optimistically universally online learnable.*

Then we also want to characterize the necessary and sufficient conditions when a process admits universal online learning when $\mathcal{H}$ has an infinite indifferent LCLL tree. A guess is the condition in the work of Hanneke & Wang (2024), which is:

*Condition* A ((Hanneke & Wang, 2024)). For a given concept class $\mathcal{H}$, there exists a countable set of experts $E = \{e_1, e_2, \dots\}$, such that $\forall \mathbb{Y}^* \in \mathrm{R}(\mathcal{H}, \mathbb{X})$, $\exists j_T \to \infty$, with $\log j_T = o(T)$, such that:

$$\mathbb{E}\left[\limsup_{T \to \infty} \min_{e_j : j \leq j_T} \frac{1}{T} \sum_{t=1}^{T} \mathbb{I}\left[e_j(X_t) \neq Y_t^*\right]\right] = 0 \tag{14}$$

Here, the expert is defined as a function that generates the label only based on $X_t$. However, this condition is not necessary. Consider that the concept class $\mathcal{H}$ contains all measurable functions. We can construct the following sequence $\mathbb{X} = \{X_t\}_{t \in \mathbb{N}}$, for every $i \in \mathbb{N}$, $X_t$ takes the same value for every $2^i - 1 \leq t \leq 2^{i+1} - 1$, but takes different value for different $i$. No matter what realizable label sequence $\mathbb{Y}^* \in \mathrm{R}(\mathcal{H}, \mathbb{X})$, this sequence admits universal online learning by memory, in other words, memorizing the history of the sequence. However, for every countable set of experts $E = \{e_1, e_2, \dots\}$, we can build the following realizable label sequence such that Condition A does not hold. For every $2^i - 1 \leq t \leq 2^{i+1} - 1$ and $j \leq 2^{2^{i+1}}$, $Y_t \neq e_j(X_t)$. It is obvious that this sequence is realizable, however, it does not satisfy Condition A, as for every $j_T \to \infty$, with $\log j_T = o(T)$, and for every $j < j_T$, we have $e_j$ makes a mistake at each round.

Therefore, the condition above is not necessary for the multiclass online learning with an unbounded label space. Thus, we left this as an open question as well.

*Open Question* 1. What is the sufficient and necessary condition of a process such that it admits universal online learning when the label space is unbounded (countably infinite)?

