# OpenReview forum: "Universal Multiclass Transductive Online Learning"
_ICML.cc/2026/Conference — ICML 2026 regular_

### Official Review · Reviewer_rKTT · 2026-03-09

**Soundness:** 3
**Presentation:** 2
**Significance:** 2
**Originality:** 2
**Overall Recommendation:** 4
**Confidence:** 2

**Summary:**

Transductive online learning is a problem in which a learner has access to a full (possibly infinite) sequence of instances in advance, and then aims to learn a mapping into (potentially infinitely many) classes. This setting differs from classical online learning in that the whole sequence (without labels) is already known at the start.

For binary classification, Hanneke et al. 2023 showed that there are three possible optimal regimes for sequence length T: O(1), O(\log T) and \Omega(T) mistakes. In this article, the authors extend this to the more general setting where the number of classes may be infinite. They show that there are the same three regimes. To do so, they use natural trees to classify

**Compliance With Llm Reviewing Policy:**

Affirmed.

**Final Justification:**

I believe that the authors obtained a result that is an interesting extension of a line of prior work. They adequately answered my questions, and I see the result as slightly more surprising now than I originally did. I still believe that the question is a bit contrived, but all in all I think this result is slightly above the bar.

**Key Questions For Authors:**

What is the surprising part about this result? That there is no area between constant or logarithmic? Did you expect a different outcome when you started studying this question?

What is the conceptual contribution over the binary case?

Could you elaborate on the examples you provide? Why are these natural choices, and how do they help the reader understand the article?

Why is an LCLL tree natural? How does it differ from the other options?

**Limitations:**

yes

**Strengths And Weaknesses:**

Soundness: Although I did not verify the mathematical claims in detail, they appear to be sound.
The definitions of the different trees are slightly confusing, but the authors provide figures to further clarify them.

Presentation: While the presentation is not
Further Comments/Nit:
"has an infinite Littlestone (LCLL) tree but has no infinite indifferent Littlestone (LCLL) tree" --> same shorthand? There is a difference between LCLL trees and Littlestone trees. Be consistent with using the names.
Figure 1 & 2 are not tidy. Things are overlapping, the subscripts are formatted inconsistently.
Section 5 is not useful in the current state. Consider expanding it a little, and sharpen up the phrasing a bit.

Significance: The authors study a novel variant of online learning. The analysis strategy seems relatively boilerplate, but requires technical care. I don't think

Originality: The question is natural but not very original. I am not familiar enough with the literature to adequately assess if there are original parts in the analysis.

---

> ### Author Rebuttal · Authors · 2026-03-30
>
> We are very grateful for the reviewer's careful review and constructive comments. We also sincerely appreciate the reviewer's encouraging score and overall assessment. We will revise our manuscript to make it clearer and less ambiguous. We will refine Figures 1 and 2 to make them more informative and intuitive. We will also revise our section 5 to sharpen the narrative and provide more information there. We also want to clarify the sentence you quoted in the weakness part here.
>
> > "has an infinite Littlestone (LCLL) tree but has no infinite indifferent Littlestone (LCLL) tree" --> same shorthand? There is a difference between LCLL trees and Littlestone trees. Be consistent with using the names
>
> We apologize for the confusing phrases. What we actually mean is that "has an infinite Littlestone (or LCLL) tree but has no infinite indifferent Littlestone (or LCLL, respectively) tree". We will change this part to ensure our words are clearer.
>
> Then, we are more than pleased to address the key questions one by one here and clarify the main contribution of this work.
>
> >What is the surprising part about this result? That there is no area between constant or logarithmic? Did you expect a different outcome when you started studying this question?
>
> The most surprising part of this result is that the characterization is not the existence of an infinite Littlestone-type tree, but instead the existence of an infinite **indifferent** Littlestone-type tree. We did expect a different characterization for this problem, which is some Littlestone-type tree. We never thought that the equivalence between the existence of an infinite Littlestone-type tree and the existence of an infinite **indifferent** Littlestone-type tree breaks when the label space is unbounded. After that, we still hope that the existence of an infinite Littlestone-type tree is the characterization. Because this means we can use the same method as previous works to construct the Gale-Stewart game. However, it turns out that we need to develop a brand new way to construct the Gale-Stewart game to ensure that the winning strategy of the adversary leads to the existence of an infinite **indifferent** Littlestone-type tree.
>
> >What is the conceptual contribution over the binary case?
>
> The main conceptual contribution, as we mentioned above, is that the **indifferent** property is important in the multiclass case and there is a way to design a Gale-Stewart game that is incompatible with this **indifferent** property.
>
> >Could you elaborate on the examples you provide? Why are these natural choices, and how do they help the reader understand the article?
>
> Because from the binary classification, we know the characterization is the existence of an infinite VCL tree, and we know that the characterization should be a Littlestone-type tree. Then there are several different extensions of VC dimension in the multiclass setting, which are LCL dimension, DS dimension, and Natarajan dimension. Therefore, a reasonable guess is that one of these three dimensions, combined with the Littlestone-type tree, is the correct answer. Then, in Section 3, we provide one examples that rule out all of these three options. The LCL tree is another possible choice of a Littlestone-type tree, but a known example rules it out as well, which is provided as Example 3.2.
>
> >Why is an LCLL tree natural? How does it differ from the other options?
>
> LCLL and DSL are both natural choices. However, we proved that they are not the correct answer; The LCLL tree equipped with the **indifferent** property, instead, is the correct answer. When we move from a binary case to a multiclass with an unbounded label space case, the **indifference** property pops out as the most important part of the combinatorial structure that characterizes the learning rates.
>
> At last, we would also like to emphasize that the proof of the lower bound of the concept class with no infinite indifferent LCLL tree in the agnostic case also significantly differs from all previous known lower bound techniques by combining Krinchine's inequality and martingale tools.
>
> We would like to thank the reviewer again for his/her time and effort.

---

> > ### Author Rebuttal · Reviewer_rKTT · 2026-04-01
> >
> > Thank you for taking time to answer my questions. I will keep my positive score.

---

> > > ### Author Response · Authors · 2026-04-04
> > >
> > > We thank the reviewer again for their time and effort.

---

### Official Review · Reviewer_mEZH · 2026-03-12

**Soundness:** 4
**Presentation:** 4
**Significance:** 4
**Originality:** 3
**Overall Recommendation:** 5
**Confidence:** 4

**Summary:**

This paper investigates the universal learning rates of transductive online multiclass learning with a countably infinite label space. The authors characterize the learnability in this setting by establishing a trichotomy of learning rates. To achieve this, they introduce a novel combinatorial structure known as the indifferent Level-Constrained-Littlestone-Littlestone (LCLL) tree. A major conceptual contribution of the work is the design of a new Gale-Stewart game that correctly captures the indifferent property of these trees

**Compliance With Llm Reviewing Policy:**

Affirmed.

**Key Questions For Authors:**

.

**Limitations:**

yes

**Strengths And Weaknesses:**

Strengths:
1. The authors derive a result that is fundamental to the learning theory community. Establishing the trichotomy of optimal rates—specifically $O(1)$, $\Theta(\log T)$, and $\Omega(T)$—for multiclass transductive online learning. This maps out the landscape of this problem beautifully.
2. Novel Combinatorial Structures: The introduction of the indifferent LCLL tree is a significant theoretical step forward for dealing with unbounded label spaces.
3. Innovative Proof Techniques: The authors note that the standard structure of the Gale-Stewart game naturally leads to an infinite Littlestone-type tree, which causes a mismatch with indifferent trees. To resolve this, they designed a game where the adversary provides all possible label sequences for an interval of instances. This conceptual contribution is a useful tool that might be used for other multiclass learning problems

---

> ### Author Rebuttal · Authors · 2026-03-30
>
> We sincerely thank the reviewer for their careful reading of our manuscript and highly positive comments. We are grateful that the reviewer found our results beautiful and our techniques innovative and potentially useful.

---

> > ### Author Rebuttal · Reviewer_mEZH · 2026-04-01
> >
> > Thanks for the authors. I will keep my score

---

> > > ### Author Response · Authors · 2026-04-04
> > >
> > > We appreciate the reviewer's time, effort, and support.

---

### Official Review · Reviewer_GxWR · 2026-03-12

**Soundness:** 4
**Presentation:** 3
**Significance:** 2
**Originality:** 3
**Overall Recommendation:** 4
**Confidence:** 3

**Summary:**

The paper studies a variant of online learning where the learner knows in advance the sequence of instances that the adversary is going to present (transductive setting). Moreover, the regret/mistake bounds depend on any fixed input sequence rather than uniformly over all input sequences (universal setting). The paper extends prior results that characterized binary setting to the multiclass setting with potentially infinite label space. The authors show that for realizable sequences of length $T$ the optimal mistake bound is either constant, $\log(T)$, or more than any sublinear function of $T$. The main contribution of the paper is deriving the logarithmic upper bound that is achieved by defining a Gale-Stewart game where the adversary proposes a sequence and a level constrained Littlestone tree and the learner must remove a branch of this tree. The learner wins at round $t$ if the proposed branch is not realizable by any hypothesis in the version space.

**Compliance With Llm Reviewing Policy:**

Affirmed.

**Final Justification:**

This is a solid theoretical paper with technical novelty.

**Key Questions For Authors:**

It was not clear to me whether the agnostic rate is a dichotomy between $\sqrt{T}$ (up to logarithmic factors) and linear regret or there are other possibilities.

**Limitations:**

yes

**Strengths And Weaknesses:**

Strengths:

The paper is theoretically solid. It proves a complete characterization of the bounds in the multiclass setting.
The presentation is generally good and the proof sketches in the main text convey the general ideas of the proofs.

Weakness:

The problem, in my opinion, is not well motivated and is interesting mostly from a theoretical view.

The proofs seem to generally follow the same prototype of prior work and the technical contributions are limited to the $\log T$ upper bound. Maybe the authors could expand more on this.

Suggestions:
I would add subsections to Section 4 to make it easier to find the results corresponding to each rate.

I think Figure 2 doesn’t fully show how your dimension is different from VCL/DSL. Maybe you could make it more clear by using specific labels.

---

> ### Author Rebuttal · Authors · 2026-03-30
>
> We thank the reviewer for the constructive suggestions. We would definitely add subsections in section 4 to make it much easier to follow for each rate. We will also change Figure 2 to make it more intuitive and different from VCL/DSL tree.
>
> We would like to explain the differences between our work and previous works here. As we mentioned in the introduction, the most surprising part is that the characterization is not the existence of a shattered infinite Littestone (or LCLL) tree, but the existence of a shattered infinite **indifferent** Littlestone (or LCLL) tree. Due to this difference, the design of the Gale-Stewart Game is completely different from all of the previous works. All of the previous works use the same type of Gale-Stewart game, in which the learner chooses the label, and the adversary chooses a sequence of instances. This type of game will automatically lead to the existence of a shattered infinite Littlestone-type tree. However, when the characterization becomes the existence of a shattered infinite **indifferent** Littlestone-type tree, that design of the Gale-Stewart game does not work, and we come up with a new design of the Gale-Stewart game, in which the winning strategy of the adversary implies the existence of a shattered infinite **indifferent** Littlestone (or LCLL) tree.
>
> As for the agnostic case, yes, you are correct. The rate of the agnostic case is a dichotomy. If the concept class has an infinite **indifferent** LCLL tree, then no learning algorithm can make $o(T)$ regret against any adversary. If the concept class has no infinite **indifferent** LCLL tree but has more than two concepts in it, there exists a learner that can have $\tilde{O}(\sqrt{T})$ regret against any adversary, but there does not exist a learner that can have $o(\sqrt{T})$ regret against any adversary. The second part is proven in Appendix C, by combining Khinchine's inequality and martingale techniques, which is also considered a brand new technique for constructing a regret lower bound in the universal online learning setting. Previous works only provide the regret lower bound for the uniform setting, which only requires an application of Khinchine's inequality.

---

> > ### Author Rebuttal · Reviewer_GxWR · 2026-04-01
> >
> > Thank you for the clarification. I maintain my positive score.

---

> > > ### Author Response · Authors · 2026-04-04
> > >
> > > We appreciate the reviewer again for their time and effort.

---

### Official Review · Reviewer_S6cf · 2026-03-15

**Soundness:** 3
**Presentation:** 3
**Significance:** 3
**Originality:** 3
**Overall Recommendation:** 4
**Confidence:** 2

**Summary:**

The paper studies universal transductive online learning for multiclass classification with countably infinite labels. It introduces the indifferent Littlestone tree and a new Level-Constrained-Littlestone-Littlestone (LCLL) tree notion, proves a realizable trichotomy separating constant, logarithmic, and impossible-sublinear regimes, gives an agnostic characterization via the absence of an infinite indifferent LCLL tree, and extends the discussion to the setting where only the instance-generating stochastic process is known. It also provides counterexamples showing that several nearby tree notions do not suffice for characterization.

Note from reviewer: I'm not the best person to review this paper, I have general knowledge about studied concepts, I personally never worked on them and I don't know many of cited works.

**Compliance With Llm Reviewing Policy:**

Affirmed.

**Key Questions For Authors:**

See weaknesses section.

**Limitations:**

Not directly, but theory seems discussed honestly and I don't see negative social impact of this work so lack of discussion is fine in this case.

**Strengths And Weaknesses:**

**Strengths**
- Sharp structural characterization of the realizable problem and the constant / logarithmic / impossible-sublinear split is clean and informative.
- I really like Section 3 with negative examples that rule out plausible but wrong alternatives.
- The proof techniques in this paper seem to be novel.
- The paper is well presented...

**Weaknesses**
- ... however, proof explanations in the main papers sometimes feel too compressed.
- It seems that the presented algorithms are not well defined, as they don't specify uniqueness or tie-breaking for argmax/argmin, etc.
$h(X_t)=y$,

---

> ### Author Rebuttal · Authors · 2026-03-30
>
> We appreciate the helpful comments made by the reviewer. We will make our proof explanations more concrete and easier to follow in the camera-ready version. We apologize that the presented algorithm might be a little bit confusing, but the way to break the tie is to just randomly choose one that satisfies the condition. The reason is that for the first algorithm, we just need the chosen label to have a weight that is larger than or equal to any other possible label, such that the total weight will decrease by at least half if we make a mistake. As for the second algorithm, if we reach the final stage of the game updates, there will only exist one label, because otherwise, we are not at the leaf of a Littlestone tree, which contradicts the fact that we reach the final stage of the game. We will explain how to break the tie in the camera-ready version to improve the accuracy of our algorithm.
> We thank the reviewer again for his/her thinkful comments and helpful suggestions.

---

> > ### Author Rebuttal · Reviewer_S6cf · 2026-04-02
> >
> > Thank you for your answer. I don't have anything to add, and I will keep my score.

---

> > > ### Author Response · Authors · 2026-04-04
> > >
> > > We thank the reviewer again for their time, effort, and positive assessments.

---

### Decision · Program_Chairs · 2026-04-30

**Decision:**

Accept (regular)

**Comment:**

Recent works have established rates of multiclass transductive online learning. This submission studies the problem under the universal PAC framework and provides a few new techniques to establish new rates. Reviewers agree that the contribution is fundamental to learning theory.